# Faster Projection-free Convex Optimization over the Spectrahedron

**Dan Garber**
Toyota Technological Institute at Chicago
dgarber@ttic.edu

## Abstract

Minimizing a convex function over the spectrahedron, i.e., the set of all $d \times d$ positive semidefinite matrices with unit trace, is an important optimization task with many applications in optimization, machine learning, and signal processing. It is also notoriously difficult to solve in large-scale since standard techniques require to compute expensive matrix decompositions. An alternative is the conditional gradient method (aka Frank-Wolfe algorithm) that regained much interest in recent years, mostly due to its application to this specific setting. The key benefit of the CG method is that it avoids expensive matrix decompositions all together, and simply requires a single eigenvector computation per iteration, which is much more efficient. On the downside, the CG method, in general, converges with an inferior rate. The error for minimizing a $\beta$-smooth function after $t$ iterations scales like $\beta/t$. This rate does not improve even if the function is also strongly convex. In this work we present a modification of the CG method tailored for the spectrahedron. The per-iteration complexity of the method is essentially identical to that of the standard CG method: only a single eigenvector computation is required. For minimizing an $\alpha$-strongly convex and $\beta$-smooth function, the *expected* error of the method after $t$ iterations is:

$$O\left(\min\{\frac{\beta}{t}, \left(\frac{\beta\sqrt{\text{rank}(\mathbf{X}^*)}}{\alpha^{1/4}t}\right)^{4/3}, \left(\frac{\beta}{\sqrt{\alpha}\lambda_{\min}(\mathbf{X}^*)t}\right)^2\}\right),$$

where $\text{rank}(\mathbf{X}^*)$, $\lambda_{\min}(\mathbf{X}^*)$ are the rank of the optimal solution and smallest non-zero eigenvalue, respectively. Beyond the significant improvement in convergence rate, it also follows that when the optimum is low-rank, our method provides better accuracy-rank tradeoff than the standard CG method. To the best of our knowledge, this is the first result that attains provably faster convergence rates for a CG variant for optimization over the spectrahedron. We also present encouraging preliminary empirical results.

## 1 Introduction

Minimizing a convex function over the set of positive semidefinite matrices with unit trace, aka the spectrahedron, is an important optimization task which lies at the heart of many optimization, machine learning, and signal processing tasks such as matrix completion [1, 13], metric learning [21, 22], kernel matrix learning [16, 9], multiclass classification [2, 23], and more.

Since modern applications are mostly of very large scale, first-order methods are the obvious choice to deal with this optimization problem. However, even these are notoriously difficult to apply, since most of the popular gradient schemes require the computation of an orthogonal projection on each iteration to enforce feasibility, which for the spectraheron, amounts to computing a full eigen-decomposition

of a real symmetric matrix. Such a decomposition requires $O(d^3)$ arithmetic operations for a $d \times d$ matrix and thus is prohibitive for high-dimensional problems. An alternative is to use first-order methods that do not require expensive decompositions, but rely only on computationally-cheap leading eigenvector computations. These methods are mostly based on the conditional gradient method, also known as the Frank-Wolfe algorithm [3, 12], which is a generic method for constrained convex optimization given an oracle for minimizing linear functions over the feasible domain. Indeed, linear minimization over the spectrahedron amounts to a single leading eigenvector computation. While the CG method has been discovered already in the 1950's [3], it has regained much interest in recent years in the machine learning and optimization communities, in particular due to its applications to semidefinite optimization and convex optimization with a nuclear norm constraint / regularization[1], e.g., [10, 13, 17, 19, 22, 2, 11]. This regained interest is not surprising: while a full eigen-decomposition for $d \times d$ matrix requires $O(d^3)$ arithmetic operations, leading eigenvecor computations can be carried out, roughly speaking, in worst-case time that is only linear in the number of non-zeros in the input matrix multiplied by either $\epsilon^{-1}$ for the popular Power Method or by $\epsilon^{-1/2}$ for the more efficient Lanczos method, where $\epsilon$ is the target accuracy. These running times improve exponentially to only depend on $\log(1/\epsilon)$ when the eigenvalues of the input matrix are well distributed [14]. Indeed, in several important machine learning applications, such as matrix completion, the CG method requires eigenvector computations of very sparse matrices [13]. Also, very recently, new eigenvector algorithms with significantly improved performance guarantees were introduced which are applicable for matrices with certain popular structure [5, 8, 20].

The main drawback of the CG method is that its convergence rate is, in general, inferior compared to projection-based gradient methods. The convergence rate for minimizing a smooth function, roughly speaking, scales only like $1/t$. In particular, this rate does not improve even when the function is also strongly convex. On the other hand, the convergence rate of optimal projection-based methods, such as Nesterov's accelerated gradient method, scales like $1/t^2$ for smooth functions, and can be improved exponentially to $\exp(-\Theta(t))$ when the objective is also strongly convex.

Very recently, several successful attempts were made to devise natural modifications of the CG method that retain the overall low per-iteration complexity, while enjoying provably faster convergence rates, usually under a strong-convexity assumption, or a slightly weaker one. These results exhibit provably-faster rates for optimization over polyhedral sets [7, 15] and strongly-convex sets [6], but do not apply to the spectrahedron. For the specific setting considered in this work, several heuristic improvements of the CG method were suggested which show promising empirical evidence, however, non of them provably improve over the rate of the standard CG method [19, 17, 4].

In this work we present a new non-trivial variant of the CG method, which, to the best of our knowledge, is the first to exhibit provably faster convergence rates for optimization over the spectrahedron, under standard smoothness and strong convexity assumptions. The per-iteration complexity of the method is essentially identical to that of the standard CG method, i.e., only a single leading eigenvector computation per iteration is required. Our method is tailored for optimization over the spectrahedron, and can be seen as a certain hybridization of the standard CG method and the projected gradient method. From a high-level view, we take advantage of the fact that solving a $\ell_2$-regularized linear problem over the set of extreme points of the spectrahedron is equivalent to linear optimization over this set, i.e., amounts to a single eigenvector computation. We then show via a novel and non-trivial analysis, that includes new decomposition concepts for positive semidefinite matrices, that such an algorithmically-cheap regularization is sufficient, in presence of strong convexity, to derive faster convergence rates.

## 2   Preliminaries and Notation

For vectors we let $\| \cdot \|$ denote the standard Euclidean norm, while for matrices we let $\| \cdot \|$ denote the spectral norm, $\|\cdot\|_F$ denote the Frobenius norm, and $\| \cdot \|_*$ denote the nuclear norm. We denote by $\mathbb{S}_d$ the space of $d \times d$ real symmetric matrices, and by $\mathcal{S}_d$ the *spectrahedron* in $\mathbb{S}_d$, i.e., $\mathcal{S}_d := \{ \mathbf{X} \in \mathbb{S}_d \, | \, \mathbf{X} \succeq 0, \mathrm{Tr}(\mathbf{X}) = 1 \}$. We let $\mathrm{Tr}(\cdot)$ and $\mathrm{rank}(\cdot)$ denote the trace and rank of a given matrix in $\mathbb{S}_d$, respectively. We let $\bullet$ denote the standard inner-product for matrices. Given a matrix $\mathbf{X} \in \mathcal{S}_d$, we let $\lambda_{\min}(\mathbf{X})$ denote the smallest non-zero eigenvalue of $\mathbf{X}$.

Given a matrix $\mathbf{A} \in \mathbb{S}_d$, we denote by $\mathbf{EV}(\mathbf{A})$ an eigenvector of $\mathbf{A}$ that corresponds to the largest (signed) eigenvalue of $\mathbf{A}$, i.e., $\mathbf{EV}(\mathbf{A}) \in \arg\max_{\mathbf{v}:\|\mathbf{v}\|=1} \mathbf{v}^\top \mathbf{A} \mathbf{v}$. Given a scalar $\xi > 0$, we also denote by $\mathbf{EV}_\xi(\mathbf{A})$ an $\xi$-approximation to the largest (in terms of eigenvalue) eigenvector of $\mathbf{A}$, i.e., $\mathbf{EV}_\xi(\mathbf{A})$ returns a unit vector $\mathbf{v}$ such that $\mathbf{v}^\top \mathbf{A} \mathbf{v} \geq \lambda_{\max}(\mathbf{A}) - \xi$.

**Definition 1.** *We say that a function $f(\mathbf{X}) : \mathbb{R}^{m \times n} \to \mathbb{R}$ is $\alpha$-strongly convex w.r.t. a norm $\|\cdot\|$, if for all $\mathbf{X}, \mathbf{Y} \in \mathbb{R}^{m \times n}$ it holds that*

$$f(\mathbf{Y}) \geq f(\mathbf{X}) + (\mathbf{Y} - \mathbf{X}) \bullet \nabla f(\mathbf{X}) + \frac{\alpha}{2}\|\mathbf{X} - \mathbf{Y}\|^2.$$

**Definition 2.** *We say that a function $f(\mathbf{X}) : \mathbb{R}^{m \times n} \to \mathbb{R}$ is $\beta$-smooth w.r.t. a norm $\|\cdot\|$, if for all $\mathbf{X}, \mathbf{Y} \in \mathbb{R}^{m \times n}$ it holds that*

$$f(\mathbf{Y}) \leq f(\mathbf{X}) + (\mathbf{Y} - \mathbf{X}) \bullet \nabla f(\mathbf{X}) + \frac{\beta}{2}\|\mathbf{X} - \mathbf{Y}\|^2.$$

The first-order optimality condition implies that for a $\alpha$-strongly convex $f$, if $\mathbf{X}^*$ is the unique minimizer of $f$ over a convex set $\mathcal{K} \subset \mathbb{R}^{m \times n}$, then for all $\mathbf{X} \in \mathcal{K}$ it holds that

$$f(\mathbf{X}) - f(\mathbf{X}^*) \geq \frac{\alpha}{2}\|\mathbf{X} - \mathbf{X}^*\|^2. \tag{1}$$

### 2.1 Problem setting

The main focus of this work is the following optimization problem:

$$\min_{\mathbf{X} \in \mathcal{S}_d} f(\mathbf{X}), \tag{2}$$

where we assume that $f(\mathbf{X})$ is both $\alpha$-strongly convex and $\beta$-smooth w.r.t. $\|\cdot\|_F$. We denote the (unique) minimizer of $f$ over $\mathcal{S}_d$ by $\mathbf{X}^*$.

## 3 Our Approach

We begin by briefly describing the conditional gradient and projected-gradient methods, pointing out their advantages and short-comings for solving Problem (2) in Subsection 3.1. We then present our new method which is a certain combination of ideas from both methods in Subsection 3.2.

### 3.1 Conditional gradient and projected gradient descent

The standard conditional gradient algorithm is detailed below in Algorithm 1.

---

**Algorithm 1** Conditional Gradient

1: input: sequence of step-sizes $\{\eta_t\}_{t \geq 1} \subset [0, 1]$
2: let $\mathbf{X}_1$ be an arbitrary matrix in $\mathcal{S}_d$
3: **for** $t = 1...$ **do**
4:     $\mathbf{v}_t \leftarrow \mathbf{EV}\left(-\nabla f(\mathbf{X}_t)\right)$
5:     $\mathbf{X}_{t+1} \leftarrow \mathbf{X}_t + \eta_t(\mathbf{v}_t \mathbf{v}_t^\top - \mathbf{X}_t)$
6: **end for**

---

Let us denote the approximation error of Algorithm 1 after $t$ iterations by $h_t := f(\mathbf{X}_t) - f(\mathbf{X}^*)$.

The convergence result of Algorithm 1 is based on the following simple observations:

$$h_{t+1} = f(\mathbf{X}_t + \eta_t(\mathbf{v}_t \mathbf{v}_t^\top - \mathbf{X}_t)) - f(\mathbf{X}^*) \tag{3}$$

$$\leq h_t + \eta_t(\mathbf{v}_t \mathbf{v}_t^\top - \mathbf{X}_t) \bullet \nabla f(\mathbf{X}_t) + \frac{\eta_t^2 \beta}{2}\|\mathbf{v}_t \mathbf{v}_t^\top - \mathbf{X}_t\|_F^2$$

$$\leq h_t + \eta_t(\mathbf{X}^* - \mathbf{X}_t) \bullet \nabla f(\mathbf{X}_t) + \frac{\eta_t^2 \beta}{2}\|\mathbf{v}_t \mathbf{v}_t^\top - \mathbf{X}_t\|_F^2 \leq (1 - \eta_t)h_t + \frac{\eta_t^2 \beta}{2}\|\mathbf{v}_t \mathbf{v}_t^\top - \mathbf{X}_t\|_F^2,$$

where the first inequality follows from the $\beta$-smoothness of $f(\mathbf{X})$, the second one follows for the optimal choice of $\mathbf{v}_t$, and the third one follows from convexity of $f(\mathbf{X})$. Unfortunately, while we

expect the error $h_t$ to rapidly converge to zero, the term $\|\mathbf{v}_t\mathbf{v}_t^\top - \mathbf{X}_t\|_F^2$ in Eq. (3), in principal, might remain as large as the diameter of $\mathcal{S}_d$, which, given a proper choice of step-size $\eta_t$, results in the well-known convergence rate of $O(\beta/t)$ [12, 10]. This consequence holds also in case $f(\mathbf{X})$ is not only smooth, but also strongly-convex.

However, in case $f$ is strongly convex, a non-trivial modification of Algorithm 1 can lead to a much faster convergence rate. In this case, it follows from Eq. (1), that on any iteration $t$, $\|\mathbf{X}_t - \mathbf{X}^*\|_F^2 \leq \frac{2}{\alpha}h_t$. Thus, if we consider replacing the choice of $\mathbf{X}_{t+1}$ in Algorithm 1 with the following update rule:

$$\mathbf{V}_t \leftarrow \arg\min_{\mathbf{V}\in\mathcal{S}_d} \mathbf{V} \bullet \nabla f(\mathbf{X}_t) + \frac{\eta_t\beta}{2}\|\mathbf{V} - \mathbf{X}_t\|_F^2, \qquad \mathbf{X}_{t+1} \leftarrow \mathbf{X}_t + \eta_t(\mathbf{V}_t - \mathbf{X}_t), \qquad (4)$$

then, following basically the same steps as in Eq. (3), we will have that

$$h_{t+1} \quad\leq\quad h_t + \eta_t(\mathbf{X}^* - \mathbf{X}_t) \bullet \nabla f(\mathbf{X}_t) + \frac{\eta_t^2\beta}{2}\|\mathbf{X}^* - \mathbf{X}_t\|_F^2 \leq \left(1 - \eta_t + \frac{\eta_t^2\beta}{\alpha}\right)h_t, \quad (5)$$

and thus by a proper choice of $\eta_t$, a linear convergence rate will be attained. Of course the issue now, is that computing $\mathbf{V}_t$ is no longer a computationally-cheap leading eigenvalue problem (in particular $\mathbf{V}_t$ is not rank-one), but requires a full eigen-decomposition of $\mathbf{X}_t$, which is much more expensive. In fact, the update rule in Eq. (4) is nothing more than the projected gradient decent method.

## 3.2 A new hybrid approach: rank one-regularized conditional gradient algorithm

At the heart of our new method is the combination of ideas from both of the above approaches: on one hand, solving a certain regularized linear problem in order to avoid the shortcomings of the CG method, i.e., slow convergence rate, and on the other hand, maintaining the simple structure of a leading eigenvalue computation that avoids the shortcoming of the computationally-expensive projected-gradient method.

Towards this end, suppose that we have an explicit decomposition of the current iterate $\mathbf{X}_t = \sum_{i=1}^k a_i\mathbf{x}_i\mathbf{x}_i^\top$, where $(a_1, a_2, ..., a_k)$ is a probability distribution over $[k]$, and each $\mathbf{x}_i$ is a unit vector. Note in particular that the standard CG method (Algorithm 1) naturally produces such an explicit decomposition of $\mathbf{X}_t$ (provided $\mathbf{X}_1$ is chosen to be rank-one). Consider now the update rule in Eq. (4), but with the additional restriction that $\mathbf{V}_t$ is rank one, i.e, $\mathbf{V}_t \leftarrow \arg\min_{\mathbf{V}\in\mathcal{S}_d, \text{rank}(\mathbf{V})=1} \mathbf{V} \bullet \nabla f(\mathbf{X}_t) + \frac{\eta_t\beta}{2}\|\mathbf{V} - \mathbf{X}_t\|_F^2$. Note that in this case it follows that $\mathbf{V}_t$ is a unit trace rank-one matrix which corresponds to the leading eigenvector of the matrix $-\nabla f(\mathbf{X}_t) + \eta_t\beta\mathbf{X}_t$. However, when $\mathbf{V}_t$ is rank-one, the regularization $\|\mathbf{V}_t - \mathbf{X}_t\|_F^2$ makes little sense in general, since unless $\mathbf{X}^*$ is rank-one, we do not expect $\mathbf{X}_t$ to be such (note however, that if $\mathbf{X}^*$ is rank one, this modification will already result in a linear convergence rate). However, we can think of solving a set of decoupled component-wise regularized problems:

$$\forall i \in [k]: \mathbf{v}_t^{(i)} \leftarrow \arg\min_{\|\mathbf{v}\|=1} \mathbf{v}^\top\nabla f(\mathbf{X}_t)\mathbf{v} + \frac{\eta_t\beta}{2}\|\mathbf{v}\mathbf{v}^\top - \mathbf{x}_i\mathbf{x}_i^\top\|_F^2 \equiv \mathbf{EV}\left(-\nabla f(\mathbf{X}_t) + \eta_t\beta\mathbf{x}_i\mathbf{x}_i^\top\right)$$

$$\mathbf{X}_{t+1} \leftarrow \sum_{i=1}^k a_i\left((1-\eta_t)\mathbf{x}_i\mathbf{x}_i^\top + \eta_t\mathbf{v}_t^{(i)}\mathbf{v}_t^{(i)\top}\right), \qquad (6)$$

where the equivalence in the first line follows since $\|\mathbf{v}\mathbf{v}^\top\|_F = 1$, and thus the minimizer of the LHS is w.l.o.g. a leading eigenvector of the matrix on the RHS. Following the lines of Eq. (3), we will now have that

$$h_{t+1} \leq h_t + \eta_t\sum_{i=1}^k a_i(\mathbf{v}_t^{(i)}\mathbf{v}_t^{(i)\top} - \mathbf{x}_i\mathbf{x}_i^\top) \bullet \nabla f(\mathbf{X}_t) + \frac{\eta_t^2\beta}{2}\|\sum_{i=1}^k a_i(\mathbf{v}_t^{(i)}\mathbf{v}_t^{(i)\top} - \mathbf{x}_i\mathbf{x}_i^\top)\|_F^2$$

$$\leq h_t + \eta_t\sum_{i=1}^k a_i(\mathbf{v}_t^{(i)}\mathbf{v}_t^{(i)\top} - \mathbf{x}_i\mathbf{x}_i^\top) \bullet \nabla f(\mathbf{X}_t) + \frac{\eta_t^2\beta}{2}\sum_{i=1}^k a_i\|\mathbf{v}_t^{(i)}\mathbf{v}_t^{(i)\top} - \mathbf{x}_i\mathbf{x}_i^\top\|_F^2$$

$$= h_t + \eta_t\mathbb{E}_{i\sim(a_1,...,a_k)}\left[(\mathbf{v}_t^{(i)}\mathbf{v}_t^{(i)\top} - \mathbf{x}_i\mathbf{x}_i^\top) \bullet \nabla f(\mathbf{X}_t) + \frac{\eta_t\beta}{2}\|\mathbf{v}_t^{(i)}\mathbf{v}_t^{(i)\top} - \mathbf{x}_i\mathbf{x}_i^\top\|_F^2\right], \quad (7)$$

where the second inequality follows from convexity of the squared Frobenius norm, and the last equality follows since $(a_1, ..., a_k)$ is a probability distribution over $[k]$.

While the approach in Eq. (6) relies only on leading eigenvector computations, the benefit in terms of potential convergence rates is not trivial, since it is not immediate that we can get non-trivial bounds for the individual distances $\|\mathbf{v}_t^{(i)}\mathbf{v}_t^{(i)\top} - \mathbf{x}_i\mathbf{x}_i^\top\|_F$. Indeed, the main novelty in our analysis is dedicated precisely to this issue. A motivation, if any, is that there might exists a decomposition of $\mathbf{X}^*$ as $\mathbf{X}^* = \sum_{i=1}^k b_i \mathbf{x}^{*(i)}\mathbf{x}^{*(i)\top}$, which is close in some sense to the decomposition of $\mathbf{X}_t$. We can then think of the regularized problem in Eq. (6), as an attempt to push each individual component $\mathbf{x}^{(i)}$ towards its corresponding component in the decomposition of $\mathbf{X}^*$, and as an overall result, bring the following iterate $\mathbf{X}_{t+1}$ closer to $\mathbf{X}^*$.

Note that Eq. (7) implicitly describes a randomized algorithm in which, instead of solving a regularized EV problem for each rank-one matrix in the decomposition of $\mathbf{X}_t$, which is expensive as this decomposition grows large with the number of iterations, we pick a single rank-one component according to its weight in the decomposition, and only update it. This directly brings us to our proposed algorithm, Algorithm 2, which is given below.

---

**Algorithm 2** Randomized Rank one-regularized Conditional Gradient

---

1: input: sequence of step-sizes $\{\eta_t\}_{t \geq 1}$, sequence of error tolerances $\{\xi_t\}_{t \geq 0}$
2: let $\mathbf{x}_0$ be an arbitrary unit vector
3: $\mathbf{X}_1 \leftarrow \mathbf{x}_1\mathbf{x}_1^\top$ such that $\mathbf{x}_1 \leftarrow \mathbf{EV}_{\xi_0}(-\nabla f(\mathbf{x}_0\mathbf{x}_0^\top))$
4: **for** $t = 1...$ **do**
5:      suppose $\mathbf{X}_t$ is given by $\mathbf{X}_t = \sum_{i=1}^k a_i\mathbf{x}_i\mathbf{x}_i^\top$, where each $\mathbf{x}_i$ is a unit vector, and $(a_1, a_2, ..., a_k)$
     is a probability distribution over $[k]$, for some integer $k$
6:      pick $i_t \in [k]$ according to the probability distribution $(a_1, a_2, ...a_k)$
7:      set a new step-size $\tilde{\eta}_t$ as follows:

$$\tilde{\eta}_t \leftarrow \begin{cases} \eta_t/2 & \text{if } a_{i_t} \geq \eta_t \\ a_{i_t} & else \end{cases}$$

8:      $\mathbf{v}_t \leftarrow \mathbf{EV}_{\xi_t}\left(-\nabla f(\mathbf{X}_t) + \eta_t \beta \mathbf{x}_{i_t}\mathbf{x}_{i_t}^\top\right)$
9:      $\mathbf{X}_{t+1} \leftarrow \mathbf{X}_t + \tilde{\eta}_t(\mathbf{v}_t\mathbf{v}_t^\top - \mathbf{x}_{i_t}\mathbf{x}_{i_t}^\top)$
10: **end for**

---

We have the following guarantee for Algorithm 2 which is the main result of this paper.

**Theorem 1.** *[Main Theorem] Consider the sequence of step-sizes $\{\eta_t\}_{t \geq 1}$ defined by $\eta_t = 18/(t + 8)$, and suppose that $\xi_0 = \beta$ and for any iteration $t \geq 1$ it holds that $\xi_t = O\left(\min\{\frac{\beta}{t}, \left(\frac{\beta\sqrt{rank(\mathbf{X}^*)}}{\alpha^{1/4}t}\right)^{4/3}, \left(\frac{\beta}{\sqrt{\alpha}\lambda_{\min}(\mathbf{X}^*)t}\right)^2\}\right)$. Then, all iterates are feasible, and*

$$\forall t \geq 1: \quad \mathbb{E}[f(\mathbf{X}_t) - f(\mathbf{X}^*)] = O\left(\min\{\frac{\beta}{t}, \left(\frac{\beta\sqrt{rank(\mathbf{X}^*)}}{\alpha^{1/4}t}\right)^{4/3}, \left(\frac{\beta}{\sqrt{\alpha}\lambda_{\min}(\mathbf{X}^*)t}\right)^2\}\right).$$

It is important to note that the step-size choice in Theorem 1 does not require any knowledge on the parameters $\alpha, \beta, \text{rank}(\mathbf{X}^*)$, and $\lambda_{\min}(\mathbf{X}^*)$. The knowledge of $\beta$ is required however for the EV computations. While it follows from Theorem 1 that the knowledge of $\alpha, \text{rank}(\mathbf{X}^*), \lambda_{\min}(\mathbf{X}^*)$ is needed to set the accuracy parameters - $\xi_t$, in practice, iterative eigenvector methods are very efficient and are much less sensitive to exact knowledge of parameters than the choice of step-size for instance.

While the eigenvalue problem in Algorithm 2 is different from the one in Algorithm 1, due to the additional term in $\mathbf{x}_{i_t}\mathbf{x}_{i_t}^\top$, the efficiency of solving both problems is essentially the same since efficient EV procedures are based on iteratively multiplying the input matrix with a vector. In particular, multiplying a vector with a rank-one matrix takes $O(d)$ time. Thus, as long as $\text{nnz}(\nabla f(\mathbf{X}_t)) = \Omega(d)$, which is highly reasonable, both EV computations run in essentially the same time.

Finally, note also that aside from the computation of the gradient direction and the leading eigenvector computation, all other operations on any iteration $t$, can be carried out in $O(d^2 + t)$ additional time.

# 4 Analysis

The complete proof of Theorem 1 and all supporting lemmas are given in full detail in the appendix. Here we only detail the two main ingredients in the analysis of Algorithm 2.

Throughout this section, given a matrix $\mathbf{Y} \in \mathcal{S}_d$, we let $\mathbf{P}_{\mathbf{Y},\tau} \in \mathbb{S}_d$ denote the projection matrix onto all eigenvectors of $\mathbf{Y}$ that correspond to eigenvalues of magnitude at least $\tau$. Similarly, we let $\mathbf{P}_{\mathbf{Y},\tau}^{\perp}$ denote the projection matrix onto the eigenvectors of $\mathbf{Y}$ that correspond to eigenvalues of magnitude smaller than $\tau$ (including eigenvectors that correspond to zero-valued eigenvalues).

## 4.1 A new decomposition for positive semidefinite matrices with locality properties

The analysis of Algorithm 2 relies heavily on a new decomposition idea of matrices in $\mathcal{S}_d$ that suggests that given a matrix $\mathbf{X}$ in the form of a convex combination of rank-one matrices: $\mathbf{X} = \sum_{i=1}^{k} \alpha_i \mathbf{x}_i \mathbf{x}_i^\top$, and another matrix $\mathbf{Y} \in \mathcal{S}_d$, roughly speaking, we can decompose $\mathbf{Y}$ as the sum of rank-one matrices, such that the components in the decomposition of $\mathbf{Y}$ are close to those in the decomposition of $\mathbf{X}$ in terms of the overall distance $\|\mathbf{X} - \mathbf{Y}\|_F$. This decomposition and corresponding property justifies the idea of solving rank-one regularized problems, as suggested in Eq. (6), and applied in Algorithm 2.

**Lemma 1.** *Let* $\mathbf{X}, \mathbf{Y} \in \mathcal{S}_d$ *such that* $\mathbf{X}$ *is given as* $\mathbf{X} = \sum_{i=1}^{k} a_i \mathbf{x}_i \mathbf{x}_i^\top$, *where each* $\mathbf{x}_i$ *is a unit vector, and* $(a_1, ..., a_k)$ *is a distribution over* $[k]$, *and let* $\tau, \gamma \in [0, 1]$ *be scalars that satisfy* $\frac{\gamma\tau}{1-\gamma} \geq \|\mathbf{X} - \mathbf{Y}\|_F$. *Then,* $\mathbf{Y}$ *can be written as* $\mathbf{Y} = \sum_{i=1}^{k} b_i \mathbf{y}_i \mathbf{y}_i^\top + \sum_{j=1}^{k} (a_j - b_j)\mathbf{W}$, *such that*

1. *each* $\mathbf{y}_i$ *is a unit vector and* $\mathbf{W} \in \mathcal{S}_d$

2. $\forall i \in [k] : 0 \leq b_i \leq a_i$ *and* $\sum_{j=1}^{k}(a_j - b_j) \leq \sqrt{rank(\mathbf{Y})}\left(\|\mathbf{Y}\mathbf{P}_{\mathbf{Y},\tau}^{\perp}\|_F + \|\mathbf{X} - \mathbf{Y}\|_F\right) + \gamma$

3. $\sum_{i=1}^{k} b_i \|\mathbf{x}_i \mathbf{x}_i^\top - \mathbf{y}_i \mathbf{y}_i^\top\|_F^2 \leq 2\sqrt{rank(\mathbf{Y})}\left(\|\mathbf{Y}\mathbf{P}_{\mathbf{Y},\tau}^{\perp}\|_F + \|\mathbf{X} - \mathbf{Y}\|_F\right)$

## 4.2 Bounding the per-iteration improvement

The main step in the proof of Theorem 1, is understanding the per-iteration improvement, as captured in Eq. (7), achievable by applying the update rule in Eq. (6), which updates on each iteration all of the rank-one components in the decomposition of the current iterate.

**Lemma 2.** *[full deterministic update] Fix a scalar* $\eta > 0$. *Let* $\mathbf{X} \in \mathcal{S}_d$ *such that* $\mathbf{X} = \sum_{i=1}^{k} a_i \mathbf{x}_i \mathbf{x}_i^\top$, *where each* $\mathbf{x}_i$ *is a unit vector, and* $(a_1, ..., a_k)$ *is a probability distribution over* $[k]$. *For any* $i \in [k]$, *let* $\mathbf{v}_i := \mathbf{EV}\left(-\nabla f(\mathbf{X}) + \eta\beta\mathbf{x}_i\mathbf{x}_i^\top\right)$. *Then, it holds that*

$$\sum_{i=1}^{k} a_i \left[(\mathbf{v}_i\mathbf{v}_i^\top - \mathbf{x}_i\mathbf{x}_i^\top) \bullet \nabla f(\mathbf{X}) + \frac{\eta\beta}{2}\|\mathbf{v}_i\mathbf{v}_i^\top - \mathbf{x}_i\mathbf{x}_i^\top\|_F^2\right] \leq -\left(f(\mathbf{X}) - f(\mathbf{X}^*)\right)$$

$$+\eta\beta \cdot \min\{1, 5\sqrt{\sqrt{\frac{2}{\alpha}}rank(\mathbf{X}^*)\sqrt{f(\mathbf{X}) - f(\mathbf{X}^*)}}, \frac{3\sqrt{2}}{\sqrt{\alpha}\lambda_{\min}(\mathbf{X}^*)}\sqrt{f(\mathbf{X}) - f(\mathbf{X}^*)}\}.$$

*proof sketch.* The proof is divided to three parts, each corresponding to a different term in the $\min$ expression in the bound in the Lemma. The first bound, at a high-level, follows from the standard conditional gradient analysis (see Eq. (3)). We continue to derive the second and third bounds.

From Lemma 1 we know we can write $\mathbf{X}^*$ in the following way:

$$\mathbf{X}^* = \sum_{i=1}^{k} b_i^* \mathbf{y}_i^* \mathbf{y}_i^{*\top} + \sum_{j=1}^{k} (a_j - b_j^*)\mathbf{W}^*, \tag{8}$$

where for all $i \in [k]$, $b_i^* \in [0, a_i]$ and $\mathbf{y}_i^*$ is a unit vector, and $\mathbf{W}^* \in \mathcal{S}_d$.

Using nothing more than Eq. (8), the optimality of $\mathbf{v}_i$ for each $i \in [k]$, and the bounds in Lemma 1, it can be shown that

$$\sum_{i=1}^{k} a_i \left[ (\mathbf{v}_i \mathbf{v}_i^\top - \mathbf{x}_i \mathbf{x}_i^\top) \bullet \nabla f(\mathbf{X}) + \frac{\eta\beta}{2} \|\mathbf{v}_i \mathbf{v}_i^\top - \mathbf{x}_i \mathbf{x}_i^\top\|_F^2 \right] \leq$$

$$(\mathbf{X}^* - \mathbf{X}) \bullet \nabla f(\mathbf{X}) + \frac{\eta\beta}{2} \sum_{i=1}^{k} b_i^* \|\mathbf{y}_i^* \mathbf{y}_i^{*\top} - \mathbf{x}_i \mathbf{x}_i^\top\|_F^2 + \eta\beta \sum_{i=1}^{k} (a_i - b_i^*) \leq$$

$$(\mathbf{X}^* - \mathbf{X}) \bullet \nabla f(\mathbf{X}) + \eta\beta \left( 2\sqrt{\mathrm{rank}(\mathbf{X}^*)} \left( \|\mathbf{X}^* \mathbf{P}_{\mathbf{X}^*,\tau}^\perp\|_F + \|\mathbf{X} - \mathbf{X}^*\|_F \right) + \gamma \right). \quad (9)$$

Now we can optimize the above bound in terms of $\tau, \gamma$. One option is to upper bound $\|\mathbf{X}^* \mathbf{P}_{\mathbf{X}^*,\tau}^\perp\|_F \leq \sqrt{\mathrm{rank}(\mathbf{X}^*)}\tau$, which together with the choice $\tau_1 = \sqrt{\frac{\|\mathbf{X}-\mathbf{X}^*\|_F}{2\mathrm{rank}(\mathbf{X}^*)}}$, $\gamma_1 = \sqrt{2\mathrm{rank}(\mathbf{X}^*)\|\mathbf{X}-\mathbf{X}^*\|_F}$, give us:

$$\text{RHS of (9)} \leq (\mathbf{X}^* - \mathbf{X}) \bullet \nabla f(\mathbf{X}) + 5\eta\beta\sqrt{\mathrm{rank}(\mathbf{X}^*)\|\mathbf{X}-\mathbf{X}^*\|_F}. \quad (10)$$

Another option, is to choose $\tau_2 = \lambda_{\min}(\mathbf{X}^*), \gamma_2 = \frac{\|\mathbf{X}-\mathbf{X}^*\|_F}{\lambda_{\min}(\mathbf{X}^*)}$ which gives us $\|\mathbf{X}^* \mathbf{P}_{\mathbf{X}^*,\tau}^\perp\|_F = 0$. This results in the bound:

$$\text{RHS of (9)} \quad \leq \quad (\mathbf{X}^* - \mathbf{X}) \bullet \nabla f(\mathbf{X}) + \frac{3\eta\beta\|\mathbf{X}-\mathbf{X}^*\|_F}{\lambda_{\min}(\mathbf{X}^*)}. \quad (11)$$

Now, using the convexity of $f$ to upper bound $(\mathbf{X}^* - \mathbf{X}) \bullet \nabla f(\mathbf{X}) \leq -(f(\mathbf{X}) - f(\mathbf{X}^*))$ and Eq. (1) in both Eq. (10) and (11), gives the second the third parts of the bound in the lemma.

$\square$

# 5 Preliminary Empirical Evaluation

We evaluate our method, along with other conditional gradient variants, on the task of matrix completion [13].

**Setting** The underlying optimization problem for the matrix completion task is the following:

$$\min_{\mathbf{Z} \in \mathcal{NB}_{d_1,d_2}(\theta)} \left\{ f(\mathbf{Z}) := \frac{1}{2} \sum_{l=1}^{n} (\mathbf{Z} \bullet \mathbf{E}_{i_l,j_l} - r_l)^2 \right\}, \quad (12)$$

where $\mathbf{E}_{i,j}$ is the indicator matrix for the entry $(i,j)$ in $\mathbb{R}^{d_1 \times d_2}$, $\{(i_l, j_l, r_l)\}_{l=1}^{n} \subset [d_1] \times [d_2] \times \mathbb{R}$, and $\mathcal{NB}_{d_1,d_2}(\theta)$ denotes the nuclear-norm ball of radius $\theta$ in $\mathbb{R}^{d_1 \times d_2}$, i.e.,

$$\mathcal{NB}_{d_1,d_2}(\theta) := \left\{ \mathbf{Z} \in \mathbb{R}^{d_1 \times d_2} \mid \|\mathbf{Z}\|_* := \sum_{i=1}^{\min\{d_1,d_2\}} \sigma_i(\mathbf{Z}) \leq \theta \right\},$$

where we let $\sigma(\mathbf{Z})$ denote the vector of singular values of $\mathbf{Z}$. . That is, our goal is to find a matrix with bounded nuclear norm (which serves as a convex surrogate for bounded rank) which matches best the partial observations given by $\{(i_l, j_l, r_l)\}_{l=1}^{n}$.

In order to transform Problem (12) to optimization over the spectrahedron, we use the reduction specified in full detail in [13], and also described in Section A in the appendix.

The objective function in Eq. (12) is known to have a smoothness parameter $\beta$ with respect to $\|\cdot\|_F$, which satisfies $\beta = O(1)$, see for instance [13]. While the objective function in Eq. (12) is not strongly convex, it is known that under certain conditions, the matrix completion problem exhibit properties very similar to strong convexity, in the sense of Eq. (1) (which is indeed the only consequence of strong convexity that we use in our analysis) [18].

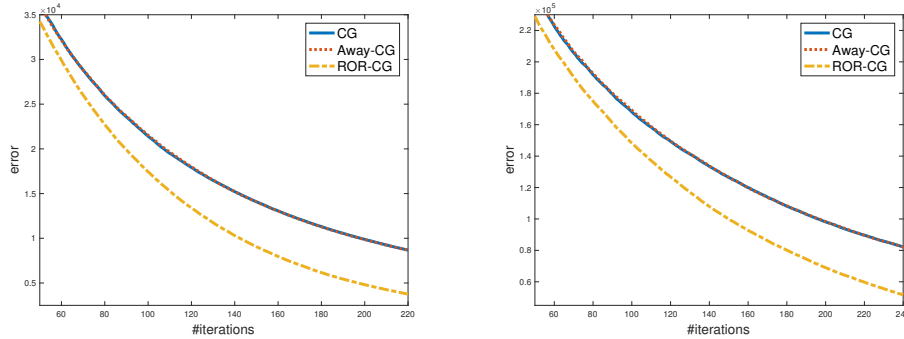

Figure 1: Comparison between conditional gradient variants for solving the matrix completion problem on the MOVIELENS100K (left) and MOVIELENS1M (right) datasets.

**Two modifications of Algorithm 2**    We implemented our rank one-regularized conditional gradient variant, Algorithm 2 (denoted ROR-CG in our figures) with two modifications. First, on each iteration $t$, instead of picking an index $i_t$ of a rank-one matrix in the decomposition of the current iterate at random according to the distribution $(a_1, a_2, ..., a_k)$, we choose it in a greedy way, i.e., we choose the rank-one component that has the largest product with the current gradient direction. While this approach is computationally more expensive, it could be easily parallelized since all dot-product computations are independent of each other. Second, after computing the eigenvector $\mathbf{v}_t$ using the step-size $\eta_t = 1/t$ (which is very close to that prescribed in Theorem 1), we apply a line-search, as detailed in [13], in order to the determine the optimal step-size given the direction $\mathbf{v}_t\mathbf{v}_t^\top - \mathbf{x}_{i_t}\mathbf{x}_{i_t}^\top$.

**Baselines**    As baselines for comparison we used the standard conditional gradient method with exact line-search for setting the step-size (denoted CG in our figures)[13], and the conditional gradient with away-steps variant, recently studied in [15] (denoted Away-CG in our figures). While the away-steps variant was studied in the context of optimization over polyhedral sets, and its formal improved guarantees apply only in that setting, the concept of away-steps still makes sense for any convex feasible set. This variant also allows the incorporation of an exact line-search procedure to choose the optimal step-size.

**Datasets**    We have experimented with two well known datasets for the matrix completion task: the MOVIELENS100K dataset for which $d_1 = 943, d_2 = 1682, n = 10^5$, and the MOVIELENS1M dataset for which $d_1 = 6040, d_2 = 3952, n \approx 10^6$. The MOVIELENS1M dataset was further sub-sampled to contain roughly half of the observations. We have set the parameter $\theta$ in Problem (12) to $\theta = 10000$ for the ML100K dataset, and $\theta = 35000$ for the ML1M dataset.

Figure 1 presents the objective (12) vs. the number of iterations executed. Each graph is the average over 5 independent experiments [2]. It can be seen that our approach indeed improves significantly over the baselines in terms of convergence rate, for the setting under consideration.

## Footnotes

[1] minimizing a convex function subject to a nuclear norm constraint is efficiently reducible to the minimization of the function over the spectrahedron, as fully detailed in [13].

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
