[Supplementary Material]

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

## A  Convex Optimization with a Nuclear Norm Constraint

An important optimization problem highly-related to Problem (2), is the problem of minimizing a convex function over the set of $d_1 \times d_2$ real-valued matrices with bounded nuclear norm, i.e,

$$\min_{\mathbf{Z} \in \mathcal{NB}_{d_1,d_2}(\theta)} f(\mathbf{Z}). \tag{13}$$

Here we let $\mathcal{NB}_{d_1,d_2}(\theta)$ denote the nuclear-norm ball of radius $\theta$ in $\mathbb{R}^{d_1 \times d_2}$, i.e.,

$$\mathcal{NB}_{d_1,d_2}(\theta) := \{\mathbf{Z} \in \mathbb{R}^{d_1 \times d_2} \mid \|\mathbf{Z}\|_* := \sum_{i=1}^{\min\{d_1,d_2\}} \sigma_i(\mathbf{Z}) \le \theta\},$$

where we let $\sigma(\mathbf{Z})$ denote the vector of singular values of $\mathbf{Z}$.

Problem (13) could be directly formulated as convex optimization over the spectrahedron. Towards this end, consider now the following convex optimization problem:

$$\min_{\mathbf{X} \in \mathcal{S}_{d_1+d_2}} \hat{f}(\mathbf{X}),$$

$$\hat{f}(\mathbf{X}) := f(2\theta \cdot \mathbf{M}_1 \mathbf{X} \mathbf{M}_2), \quad \mathbf{M}_1 := (\begin{array}{cc} \mathbf{I}_{d_1} & \mathbf{0}_{d_1 \times d_2} \end{array}), \quad \mathbf{M}_1 := \left(\begin{array}{c} \mathbf{0}_{d_1 \times d_2} \\ \mathbf{I}_{d_2} \end{array}\right).$$

The following Lemma, whose proof can be found in [13], shows the equivalence between the two problems.

**Lemma 3.** *Let $\mathbf{X} \in \mathcal{S}_{d_1+d_2}$ such that $\hat{f}(\mathbf{X}) - \hat{f}(\mathbf{X}^*) = \epsilon$, for some $\epsilon > 0$, where $\mathbf{X}^*$ is the minimizer of $\hat{f}$ over $\mathcal{S}_{d_1+d_2}$. Consider the following factorization of $\mathbf{X}$:*

$$\mathbf{X} = \left(\begin{array}{cc} \mathbf{X}_1 & \mathbf{X}_2 \\ \mathbf{X}_2^\top & \mathbf{X}_3 \end{array}\right),$$

*where $\mathbf{X}_1$ is $d_1 \times d_1$, $\mathbf{X}_2$ is $d_1 \times d_2$, and $\mathbf{X}_3$ is $d_2 \times d_2$. Define $\mathbf{Z} := 2\theta \cdot \mathbf{X}_2$. Then it follows that $\mathbf{Z} \in \mathcal{NB}_{d_1,d_2}(\theta)$, and $f(\mathbf{Z}) - f(\mathbf{Z}^*) = \epsilon$, where $\mathbf{Z}^*$ is the minimizer of $f$ over $\mathcal{NB}_{d_1,d_2}(\theta)$.*

## B  Omitted Lemmas and Proofs from Subsection 4.1

Here we give a complete proof of Lemma 1. Before we can prove the lemma , we first need two more technical lemmas.

**Lemma 4.** *Let $\mathbf{X}, \mathbf{Y} \in \mathcal{S}_d$. Let $\tau, \gamma \in [0,1]$ be scalars that satisfy $\frac{\gamma\tau}{1-\gamma} \ge \|\mathbf{X} - \mathbf{Y}\|_F$. Then it holds that $\mathbf{Y} \succeq (1-\gamma)\mathbf{P}_{\mathbf{Y},\tau} \mathbf{X} \mathbf{P}_{\mathbf{Y},\tau}$.*

*Proof.* Given a vector $\mathbf{w} \in \mathbb{R}^d$ let us write it as $\mathbf{w} = \mathbf{w}^+ + \mathbf{w}^-$ where $\mathbf{w}^+ = \mathbf{P}_{\mathbf{Y},\tau}\mathbf{w}$ and $\mathbf{w}^- = \mathbf{P}_{\mathbf{Y},\tau}^\perp \mathbf{w} = \mathbf{w} - \mathbf{w}^+$.

It holds that

$$\begin{aligned} \mathbf{w}^\top \mathbf{Y} \mathbf{w} &= \mathbf{w}^{+\top}\mathbf{Y}\mathbf{w}^+ + \mathbf{w}^{-\top}\mathbf{Y}\mathbf{w}^- + 2\mathbf{w}^{-\top}\mathbf{Y}\mathbf{w}^+ \\ &= \mathbf{w}^{+\top}\mathbf{Y}\mathbf{w}^+ + \mathbf{w}^{-\top}\mathbf{Y}\mathbf{w}^- + 2\mathbf{w}^\top \mathbf{P}_{\mathbf{Y},\tau}^\perp \mathbf{Y} \mathbf{P}_{\mathbf{Y},\tau}\mathbf{w} \\ &\ge \mathbf{w}^{+\top}\mathbf{Y}\mathbf{w}^+, \end{aligned} \tag{14}$$

where the inequality follows since $\mathbf{P}_{\mathbf{Y},\tau}^\perp \mathbf{Y} \mathbf{P}_{\mathbf{Y},\tau} = 0$ and $\mathbf{Y}$ is positive semidefinite.

Similarly, since $\mathbf{P}_{\mathbf{Y},\tau}\mathbf{w}^- = 0$, we have that

$$\mathbf{w}^{-\top}\mathbf{P}_{\mathbf{Y},\tau}\mathbf{X}\mathbf{P}_{\mathbf{Y},\tau}\mathbf{w}^- = \mathbf{w}^{-\top}\mathbf{P}_{\mathbf{Y},\tau}\mathbf{X}\mathbf{P}_{\mathbf{Y},\tau}\mathbf{w}^+ = \mathbf{w}^{+\top}\mathbf{P}_{\mathbf{Y},\tau}\mathbf{X}\mathbf{P}_{\mathbf{Y},\tau}\mathbf{w}^- = 0. \tag{15}$$

Note also that

$$\mathbf{w}^{+\top}\mathbf{P}_{\mathbf{Y},\tau}\mathbf{X}\mathbf{P}_{\mathbf{Y},\tau}\mathbf{w}^+ = \mathbf{w}^{+\top}\mathbf{X}\mathbf{w}^+. \tag{16}$$

Thus, we have that

$$
\begin{aligned}
\mathbf{w}^\top\left[(1-\gamma)\mathbf{P_{Y,\tau}}\mathbf{X}\mathbf{P_{Y,\tau}}\right]\mathbf{w} &= (1-\gamma)\mathbf{w}^{+\top}\mathbf{P}_{Y,\tau}\mathbf{X}\mathbf{P}_{Y,\tau}w^+ \\
&= (1-\gamma)\mathbf{w}^{+\top}\mathbf{X}\mathbf{w}^+ \\
&\leq (1-\gamma)\left(\mathbf{w}^{+\top}\mathbf{Y}\mathbf{w}^+ + \|\mathbf{X}-\mathbf{Y}\|_F\cdot\|\mathbf{w}^+\|^2\right) \\
&\leq \mathbf{w}^\top\mathbf{Y}\mathbf{w} + (1-\gamma)\|\mathbf{X}-\mathbf{Y}\|_F\cdot\|\mathbf{w}^+\|^2 - \gamma\mathbf{w}^{+\top}\mathbf{Y}\mathbf{w}^+ \\
&\leq \mathbf{w}^\top\mathbf{Y}\mathbf{w} + (1-\gamma)\|\mathbf{X}-\mathbf{Y}\|_F\cdot\|\mathbf{w}^+\|^2 - \gamma\tau\|\mathbf{w}^+\|^2,
\end{aligned}
$$

where the first equality follows from Eq. (15), the second equality follows from Eq. (16), the first inequality follows from the Cauchy-Schwarz ineq., the second inequality follows from Eq. (14), and the last inequality follows from the definitions of $\mathbf{w}^+$ and $\tau$.

Thus, we can see that if $\frac{\gamma\tau}{1-\gamma} \geq \|\mathbf{X}-\mathbf{Y}\|_F$, the lemma follows.

$\square$

**Lemma 5.** *Let* $\mathbf{X}, \mathbf{Y} \in \mathcal{S}_d$ *and suppose* $\mathbf{X}$ *is given in the form* $\mathbf{X} = \sum_{i=1}^k a_i \mathbf{x}_i \mathbf{x}_i^\top$ *where each* $\mathbf{x}_i$ *is a unit vector, and the weights* $(a_1, ..., a_k)$ *are a distribution over* $[k]$*. Let* $\mathbf{P} \in \mathbb{S}_d$ *be a projection matrix onto a subset of the eigenvectors of* $\mathbf{Y}$*, and define for any* $i \in [k]$*,* $\tilde{\mathbf{x}}_i := \mathbf{P}\mathbf{x}_i$*. Then, it holds that*

$$
\sum_{i=1}^k a_i(1 - \|\tilde{\mathbf{x}}_i\|^2) \leq \sqrt{rank(\mathbf{Y})}\|\mathbf{Y}-\mathbf{P}\mathbf{X}\mathbf{P}\|_F.
$$

*Proof.* Let us write the eigen-decomposition of $\mathbf{Y}$ as $\mathbf{Y} = \sum_{j=1}^{\text{rank}(\mathbf{Y})} \lambda_j \mathbf{v}_j \mathbf{v}_j^\top$. Using simple algebraic manipulations we have that

$$
\begin{aligned}
\|\mathbf{Y}-\mathbf{P}\mathbf{X}\mathbf{P}\|_F^2 &\geq \sum_{j=1}^{\text{rank}(\mathbf{Y})}\left((\mathbf{Y}-\mathbf{P}\mathbf{X}\mathbf{P})\bullet\mathbf{v}_j\mathbf{v}_j^\top\right)^2 = \sum_{j=1}^{\text{rank}(\mathbf{Y})}\left(\lambda_j - \sum_{i=1}^k a_i \mathbf{v}_j^\top \mathbf{P}\mathbf{x}_i\mathbf{x}_i^\top\mathbf{P}\mathbf{v}_j\right)^2 \\
&= \sum_{j=1}^{\text{rank}(\mathbf{Y})}\left(\lambda_j - \sum_{i=1}^k a_i(\mathbf{v}_j^\top\mathbf{P}\mathbf{x}_i)^2\right)^2 \\
&\geq \frac{1}{\text{rank}(\mathbf{Y})}\left(\sum_{j=1}^{\text{rank}(\mathbf{Y})}\left(\lambda_j - \sum_{i=1}^k a_i(\mathbf{v}_j^\top\mathbf{P}\mathbf{x}_i)^2\right)\right)^2 \\
&= \frac{1}{\text{rank}(\mathbf{Y})}\left(1 - \sum_{j=1}^{\text{rank}(\mathbf{Y})}\sum_{i=1}^k a_i(\mathbf{v}_j^\top\mathbf{P}\mathbf{x}_i)^2\right)^2 \\
&= \frac{1}{\text{rank}(\mathbf{Y})}\left(\sum_{i=1}^k a_i\left(1 - \sum_{j=1}^{\text{rank}(\mathbf{Y})}(\mathbf{v}_j^\top\mathbf{P}\mathbf{x}_i)^2\right)\right)^2 \\
&= \frac{1}{\text{rank}(\mathbf{Y})}\left(\sum_{i=1}^k a_i\left(1 - \|\tilde{\mathbf{x}}_i\|^2\right)\right)^2,
\end{aligned}
$$

where the first inequality follows since $\{\mathbf{v}_j\mathbf{v}_j^\top\}_{j=1}^{\text{rank}(\mathbf{Y})}$ is a subset of an orthonormal basis for $\mathbb{R}^{d\times d}$, the second inequality follows since for any finite set of reals $\{r_i\}_{i=1}^k$ it holds that $\sum_{i=1}^k r_i^2 \geq (\sum_{i=1}^k r_i)^2/k$, and the third and forth equalities hold since $\sum_{j=1}^{\text{rank}(\mathbf{Y})}\lambda_j = \sum_{i=1}^k a_i = 1$.

Thus we have that

$$
\sum_{i=1}^k a_i(1 - \|\tilde{\mathbf{x}}_i\|^2) \leq \sqrt{\text{rank}(\mathbf{Y})}\|\mathbf{Y}-\mathbf{P}\mathbf{X}\mathbf{P}\|_F,
$$

which gives the bound in the lemma.

$\square$

We can now prove Lemma 1.

*Proof.* For each $i \in [k]$ let $\tilde{\mathbf{x}}_i = \mathbf{P}_{\mathbf{Y},\tau}\mathbf{x}_i$. It follows from Lemma 4 that as long as $\frac{\gamma\tau}{1-\gamma} \geq \|\mathbf{X} - \mathbf{Y}\|_F$, it holds that

$$\mathbf{Y} \succeq \sum_{i=1}^{k} a_i(1-\gamma)\tilde{\mathbf{x}}_i\tilde{\mathbf{x}}_i^\top.$$

Since $\mathbf{Y} \in \mathcal{S}_d$ and $\mathrm{Tr}\left(\sum_{i=1}^{k} a_i(1-\gamma)\tilde{\mathbf{x}}_i\tilde{\mathbf{x}}_i^\top\right) = \sum_{i=1}^{k} a_i(1-\gamma)\|\tilde{\mathbf{x}}_i\|^2$, it follows that $\mathbf{Y}$ can be written as:

$$\mathbf{Y} = \sum_{i=1}^{k} a_i(1-\gamma)\tilde{\mathbf{x}}_i\tilde{\mathbf{x}}_i^\top + \left(\sum_{j=1}^{k} a_j\left(1-(1-\gamma)\|\tilde{\mathbf{x}}_j\|^2\right)\right)\mathbf{W},$$

where $\mathbf{W} \in \mathcal{S}_d$.

Let us now define $\mathbf{y}_i := \frac{\tilde{\mathbf{x}}_i}{\|\tilde{\mathbf{x}}_i\|}$ and $b_i := a_i(1-\gamma)\|\tilde{\mathbf{x}}_i\|^2$. Then indeed it follows that

$$\mathbf{Y} = \sum_{i=1}^{k} b_i\mathbf{y}_i\mathbf{y}_i^\top + \sum_{j=1}^{k}(a_j - b_j)\mathbf{W}.$$

We are going to apply Lemma 5 to derive the bounds listed in the lemma. As a first step, we need to bound the distance $\|\mathbf{Y} - \mathbf{P}_{\mathbf{Y},\tau}\mathbf{X}\mathbf{P}_{\mathbf{Y},\tau}\|_F$.

$$
\begin{aligned}
\|\mathbf{Y} - \mathbf{P}_{\mathbf{Y},\tau}\mathbf{X}\mathbf{P}_{\mathbf{Y},\tau}\|_F &\leq \|\mathbf{Y} - \mathbf{P}_{\mathbf{Y},\tau}\mathbf{Y}\mathbf{P}_{\mathbf{Y},\tau}\|_F + \|\mathbf{P}_{\mathbf{Y},\tau}\mathbf{X}\mathbf{P}_{\mathbf{Y},\tau} - \mathbf{P}_{\mathbf{Y},\tau}\mathbf{Y}\mathbf{P}_{\mathbf{Y},\tau}\|_F \\
&\leq \|\mathbf{Y}\mathbf{P}_{\mathbf{Y},\tau}^\perp\|_F + \|\mathbf{X} - \mathbf{Y}\|_F, \quad\quad (17)
\end{aligned}
$$

where the bound on $\|\mathbf{Y} - \mathbf{P}_{\mathbf{Y},\tau}\mathbf{Y}\mathbf{P}_{\mathbf{Y},\tau}\|_F$ follows from the definition of $\mathbf{P}_{\mathbf{Y},\tau}^\perp$, and the bound on $\|\mathbf{P}_{\mathbf{Y},\tau}\mathbf{X}\mathbf{P}_{\mathbf{Y},\tau} - \mathbf{P}_{\mathbf{Y},\tau}\mathbf{Y}\mathbf{P}_{\mathbf{Y},\tau}\|_F$ follows from the inequality $\|\mathbf{A}\mathbf{B}\|_F \leq \|\mathbf{A}\| \cdot \|\mathbf{B}\|_F$ (applied twice, each time with $\mathbf{A} = \mathbf{P}_{\mathbf{Y},\tau}$, and recalling that $\|\mathbf{P}_{\mathbf{Y},\tau}\| = 1$).

By definition of $\{b_i\}_{i\in[k]}$ it holds that

$$
\begin{aligned}
\sum_{i=1}^{k}(a_i - b_i) &= \sum_{i=1}^{k} a_i(1-(1-\gamma)\|\tilde{\mathbf{x}}_i\|^2) \leq \sum_{i=1}^{k} a_i(1-\|\tilde{\mathbf{x}}_i\|^2) + \gamma \\
&\leq \sqrt{\mathrm{rank}(\mathbf{Y})}\left(\|\mathbf{Y}\mathbf{P}_{\mathbf{Y},\tau}^\perp\|_F + \|\mathbf{X} - \mathbf{Y}\|_F\right) + \gamma,
\end{aligned}
$$

where the last inequality follows from Lemma 5 and the bound in Eq. (17).

We continue to upper-bound $\sum_{i=1}^{k} b_i\|\mathbf{x}_i\mathbf{x}_i^\top - \mathbf{y}_i\mathbf{y}_i^\top\|_F^2$:

$$
\begin{aligned}
\sum_{i=1}^{k} b_i\|\mathbf{x}_i\mathbf{x}_i^\top - \mathbf{y}_i\mathbf{y}_i^\top\|_F^2 &\leq \sum_{i=1}^{k} a_i\|\mathbf{x}_i\mathbf{x}_i^\top - \mathbf{y}_i\mathbf{y}_i^\top\|_F^2 \\
&= 2\sum_{i=1}^{k} a_i(1-(\mathbf{x}_i^\top\mathbf{y}_i)^2) \\
&= 2\sum_{i=1}^{k} a_i\left(1-\left(\frac{\mathbf{x}_i^\top\tilde{\mathbf{x}}_i}{\|\tilde{\mathbf{x}}_i\|}\right)^2\right) \\
&= 2\sum_{i=1}^{k} a_i(1-\|\tilde{\mathbf{x}}_i\|^2) \\
&\leq 2\sqrt{\mathrm{rank}(\mathbf{Y})}\left(\|\mathbf{Y}\mathbf{P}_{\mathbf{Y},\tau}^\perp\|_F + \|\mathbf{X} - \mathbf{Y}\|_F\right),
\end{aligned}
$$

where the last inequality follows again from the application of Lemma 5 and the bound in Eq. (17). $\quad\square$

# C Omitted Lemmas and Proofs from Subsection 4.2

We now turn to analyze the per-iteration improvement of Algorithm 2. We start by first analyzing a deterministic, and much less efficient, version that updates all of the rank-one components on each iteration $t$, as suggested in Eq. (6). This is done in Lemma 2, whose complete proof is given here. Then in Lemma 7, we apply Lemma 2, to analyze the randomized step of Algorithm 2. However, first we need a simple observation regarding Algorithm 2, that shows that it can always take sufficiently large step-sizes, i.e., step-size of magnitude at least $\eta_t/2$ on iteration $t$.

**Observation 1.** *In case the input sequence of step-sizes in Algorithm 2 - $\{\eta_t\}_{t\geq 1}$, is monotonically non-increasing and $\eta_t \in [0, 2]$ for all $t \geq 1$, it follows that on each iteration $t$ of the algorithm, the iterate $\mathbf{X}_t$ admits an explicitly-given factorization into a convex sum of rank-one matrices, as described in the algorithm, such that for every rank-one coefficient $a_i$, it holds that $a_i \geq \eta_t/2$.*

*Proof.* The proof is by a simple induction. Since $\mathbf{X}_1$ is just a rank-one matrix, it follows that the corresponding coefficient in the convex sum is $a_1 = 1$. Thus, for any $\eta_1 \in [0, 2]$ it indeed follows that $a_1 \geq \eta_1/2$. Assume now that the induction holds for time $t \geq 1$. On time $t$ we choose a coefficient $a_{i_t}$ and move a mass of $\tilde{\eta}_t$ from it to a new rank-one matrix $\mathbf{y}_t \mathbf{y}_t^\top$, and all other coefficients remain unchanged. Since we assume that the step-size sequence is monotonically non-increasing, it directly follows that the induction step holds for all unchanged coefficients. Regarding the affected coefficients $a_{i_t}$ and the coefficient of the new rank-one matrix, we consider two cases. First, if $a_{i_t} \geq \eta_t$ then by the definition of $\tilde{\eta}_t$ we have that the mass of the new coefficient is going to be exactly $\eta_t/2$ and the mass of the old coefficient is going to be $a_{i_t} - \eta_t/2 \geq \eta_t/2$, and thus the induction holds. In the other case, we have that $\tilde{\eta}_t = a_{i_t} < \eta_t$. By the induction hypothesis we know that $a_{i_t} \geq \eta_t/2 \geq \eta_{t+1}/2$. Since we are moving now all the mass from $a_{i_t}$ to the new rank-one matrix, it follows that its weight is also going to be at least $\eta_{t+1}/2$, and thus the induction follows. $\square$

We now restate Lemma 2 and give its complete proof. In this version of the lemma we also take into consideration the approximation errors in the eigenvector computations.

**Lemma 6.** *[full deterministic update] Fix a scalar $\eta > 0$. Let $\mathbf{X} \in \mathcal{S}_d$ such that $\mathbf{X} = \sum_{i=1}^k a_i \mathbf{x}_i \mathbf{x}_i^\top$, where each $\mathbf{x}_i$ is a unit vector, and $(a_1, ..., a_k)$ is a probability distribution over $[k]$. For any $i \in [k]$, let*

$$\mathbf{v}_i := \mathbf{EV}_\xi \left( -\nabla f(\mathbf{X}) + \eta \beta \mathbf{x}_i \mathbf{x}_i^\top \right), \tag{18}$$

*for some parameter $\xi > 0$. Then, it holds that*

$$\sum_{i=1}^k a_i \left[ (\mathbf{v}_i \mathbf{v}_i^\top - \mathbf{x}_i \mathbf{x}_i^\top) \bullet \nabla f(\mathbf{X}) + \frac{\eta \beta}{2} \|\mathbf{v}_i \mathbf{v}_i^\top - \mathbf{x}_i \mathbf{x}_i^\top\|_F^2 \right] \leq - (f(\mathbf{X}) - f(\mathbf{X}^*))$$

$$+ \eta \beta \cdot \min\{1, 5\sqrt{\sqrt{\frac{2}{\alpha}} rank(\mathbf{X}^*) \sqrt{f(\mathbf{X}) - f(\mathbf{X}^*)}}, \frac{3\sqrt{2}}{\sqrt{\alpha}\lambda_{\min}(\mathbf{X}^*)} \sqrt{f(\mathbf{X}) - f(\mathbf{X}^*)}\} + \xi.$$

*Proof.* For the sake of clarity, throughout the proof we treat each $\mathbf{v}_i$ as the result of an exact eigenvector computation, i.e., we assume $\xi = 0$, and at the end we discuss the effect of the approximation error in the computation of $\mathbf{v}_i$.

Let $\mathbf{w}^* \in \arg\min_{\mathbf{w}:\|\mathbf{w}\|=1} \mathbf{w}^\top \nabla f(\mathbf{X})\mathbf{w}$. Using the optiamlity of $\mathbf{v}_i$, and the fact that for every $i \in [k]$, both $\mathbf{v}_i$ and $\mathbf{x}_i$ are unit vectors, we have that

$$\sum_{i=1}^{k} a_i \left[ (\mathbf{v}_i \mathbf{v}_i^\top - \mathbf{x}_i \mathbf{x}_i^\top) \bullet \nabla f(\mathbf{X}) + \frac{\eta\beta}{2} \|\mathbf{v}_i \mathbf{v}_i^\top - \mathbf{x}_i \mathbf{x}_i^\top\|_F^2 \right] \le$$

$$\sum_{i=1}^{k} a_i \left[ (\mathbf{w}^* \mathbf{w}^{*\top} - \mathbf{x}_i \mathbf{x}_i^\top) \bullet \nabla f(\mathbf{X}) + \frac{\eta\beta}{2} \|\mathbf{w}^* \mathbf{w}^{*\top} - \mathbf{x}_i \mathbf{x}_i^\top\|_F^2 \right] \le$$

$$\sum_{i=1}^{k} a_i \left[ (\mathbf{w}^* \mathbf{w}^{*\top} - \mathbf{x}_i \mathbf{x}_i^\top) \bullet \nabla f(\mathbf{X}) + \eta\beta \right] =$$

$$(\mathbf{w}^* \mathbf{w}^{*\top} - \mathbf{X}) \bullet \nabla f(\mathbf{X}) + \eta\beta \le (\mathbf{X}^* - \mathbf{X}) \bullet \nabla f(\mathbf{X}) + \eta\beta \le -(f(\mathbf{X}) - f(\mathbf{X}^*)) + \eta\beta, \tag{19}$$

where the third inequality follows from the optimality of $\mathbf{w}^*$ (note that the minimizer of a linear function $\mathbf{A} \bullet \mathbf{X}$ when $\mathbf{X} \in \mathcal{S}_d$ and $\mathbf{A} \in \mathbb{S}_d$, is w.l.o.g. a rank-one matrix corresponding to a leading eigenvector of $\mathbf{A}$), and the last inequality follows from the convexity of $f$. Thus, Eq. (19) gives us the first part of the bound stated in the lemma. We now move on the prove the second part.

From Lemma 1 we know we can write $\mathbf{X}^*$ in the following way:

$$\mathbf{X}^* = \sum_{i=1}^{k} b_i^* \mathbf{y}_i^* \mathbf{y}_i^{*\top} + \sum_{j=1}^{k} (a_j - b_j^*) \mathbf{W}^*, \tag{20}$$

where for all $i \in [k]$, $b_i^* \in [0, a_i]$ and $\mathbf{y}_i^*$ is a unit vector, and $\mathbf{W}^* \in \mathcal{S}_d$.

Using again the optimality of $\mathbf{v}_i$ for each $i \in [k]$, we have that

$$\sum_{i=1}^{k} a_i \left[ (\mathbf{v}_i \mathbf{v}_i^\top - \mathbf{x}_i \mathbf{x}_i^\top) \bullet \nabla f(\mathbf{X}) + \frac{\eta\beta}{2} \|\mathbf{v}_i \mathbf{v}_i^\top - \mathbf{x}_i \mathbf{x}_i^\top\|_F^2 \right] \le$$

$$\sum_{i=1}^{k} a_i \cdot \min \{ (\mathbf{y}_i^* \mathbf{y}_i^{*\top} - \mathbf{x}_i \mathbf{x}_i^\top) \bullet \nabla f(\mathbf{X}) + \frac{\eta\beta}{2} \|\mathbf{y}_i^* \mathbf{y}_i^{*\top} - \mathbf{x}_i \mathbf{x}_i^\top\|_F^2,$$

$$(\mathbf{w}^* \mathbf{w}^{*\top} - \mathbf{x}_i \mathbf{x}_i^\top) \bullet \nabla f(\mathbf{X}) + \frac{\eta\beta}{2} \|\mathbf{w}^* \mathbf{w}^{*\top} - \mathbf{x}_i \mathbf{x}_i^\top\|_F^2 \} \le$$

$$\sum_{i=1}^{k} b_i^* \left[ (\mathbf{y}_i^* \mathbf{y}_i^{*\top} - \mathbf{x}_i \mathbf{x}_i^\top) \bullet \nabla f(\mathbf{X}) + \frac{\eta\beta}{2} \|\mathbf{y}_i^* \mathbf{y}_i^{*\top} - \mathbf{x}_i \mathbf{x}_i^\top\|_F^2 \right]$$

$$+ \sum_{i=1}^{k} (a_i - b_i^*) \left[ (\mathbf{w}^* \mathbf{w}^{*\top} - \mathbf{x}_i \mathbf{x}_i^\top) \bullet \nabla f(\mathbf{X}) + \frac{\eta\beta}{2} \|\mathbf{w}^* \mathbf{w}^{*\top} - \mathbf{x}_i \mathbf{x}_i^\top\|_F^2 \right]$$

$$\le \sum_{i=1}^{k} b_i^* \left[ (\mathbf{y}_i^* \mathbf{y}_i^{*\top} - \mathbf{x}_i \mathbf{x}_i^\top) \bullet \nabla f(\mathbf{X}) + \frac{\eta\beta}{2} \|\mathbf{y}_i^* \mathbf{y}_i^{*\top} - \mathbf{x}_i \mathbf{x}_i^\top\|_F^2 \right]$$

$$+ \sum_{i=1}^{k} (a_i - b_i^*) \left[ (\mathbf{W}^* - \mathbf{x}_i \mathbf{x}_i^\top) \bullet \nabla f(\mathbf{X}) + \frac{\eta\beta}{2} \|\mathbf{w}^* \mathbf{w}^{*\top} - \mathbf{x}_i \mathbf{x}_i^\top\|_F^2 \right], \tag{21}$$

where the second inequality follows since $\min\{a, b\} \le \lambda a + (1 - \lambda)b$ for any $a, b \in \mathbb{R}$, $\lambda \in [0, 1]$, and the third inequality follows from the optimality of $\mathbf{w}^*$. Using Eq. (20) we have that

$$
\begin{aligned}
\text{RHS of (21)} \quad \le \quad & (\mathbf{X}^* - \mathbf{X}) \bullet \nabla f(\mathbf{X}) + \sum_{i=1}^{k} b_i^* \frac{\eta\beta}{2} \|\mathbf{y}_i^* \mathbf{y}_i^{*\top} - \mathbf{x}_i \mathbf{x}_i^\top\|_F^2 \\
& + \sum_{i=1}^{k} (a_i - b_i^*) \frac{\eta\beta}{2} \|\mathbf{w}^* \mathbf{w}^{*\top} - \mathbf{x}_i \mathbf{x}_i^\top\|_F^2 \\
\le \quad & (\mathbf{X}^* - \mathbf{X}) \bullet \nabla f(\mathbf{X}) + \frac{\eta\beta}{2} \sum_{i=1}^{k} b_i^* \|\mathbf{y}_i^* \mathbf{y}_i^{*\top} - \mathbf{x}_i \mathbf{x}_i^\top\|_F^2 + \eta\beta \sum_{i=1}^{k} (a_i - b_i^*) \\
\le \quad & (\mathbf{X}^* - \mathbf{X}) \bullet \nabla f(\mathbf{X}) + \eta\beta \sqrt{\text{rank}(\mathbf{X}^*)} \left( \sqrt{\text{rank}(\mathbf{X}^*)}\tau + \|\mathbf{X} - \mathbf{X}^*\|_F \right) \\
& + \eta\beta \left( \sqrt{\text{rank}(\mathbf{X}^*)} \left( \sqrt{\text{rank}(\mathbf{X}^*)}\tau + \|\mathbf{X} - \mathbf{X}^*\|_F \right) + \gamma \right) \\
= \quad & (\mathbf{X}^* - \mathbf{X}) \bullet \nabla f(\mathbf{X}) + \eta\beta \left( 2\sqrt{\text{rank}(\mathbf{X}^*)} \left( \|\mathbf{X}^* \mathbf{P}_{\mathbf{X}^*, \tau}^\perp\|_F + \|\mathbf{X} - \mathbf{X}^*\|_F \right) + \gamma \right),
\end{aligned}
\tag{22}
$$

where the last inequality follows from plugging the bounds in Lemma 1 and holds for any $\tau, \gamma \in [0, 1]$ such that $\frac{\tau\gamma}{1-\gamma} \ge \|\mathbf{X} - \mathbf{X}^*\|_F$.

Now we can optimize the above bound in terms $\tau, \gamma$ under the constraint that $\frac{\gamma\tau}{1-\gamma} \ge \|\mathbf{X} - \mathbf{X}^*\|_F$. One option is to upper bound $\|\mathbf{X}^* \mathbf{P}_{\mathbf{X}^*, \tau}^\perp\|_F \le \sqrt{\text{rank}(\mathbf{X}^*)}\tau$, which gives us

$$
\text{RHS of (21)} \quad \le \quad (\mathbf{X}^* - \mathbf{X}) \bullet \nabla f(\mathbf{X}) + \eta\beta \left( 2\sqrt{\text{rank}(\mathbf{X}^*)} \left( \sqrt{\text{rank}(\mathbf{X}^*)}\tau + \|\mathbf{X} - \mathbf{X}^*\|_F \right) + \gamma \right).
$$

We can then set:

$$
\tau_1 = \sqrt{\frac{\|\mathbf{X} - \mathbf{X}^*\|_F}{2\text{rank}(\mathbf{X}^*)}}, \qquad \gamma_1 = \sqrt{2\text{rank}(\mathbf{X}^*)\|\mathbf{X} - \mathbf{X}^*\|_F},
$$

as long as $\|\mathbf{X} - \mathbf{X}^*\|_F \le \frac{1}{2\text{rank}(\mathbf{X}^*)}$, which gives us:

$$
\text{RHS of (21)} \quad \le \quad (\mathbf{X}^* - \mathbf{X}) \bullet \nabla f(\mathbf{X}) + 2\eta\beta \sqrt{\text{rank}(\mathbf{X}^*)} \left( \sqrt{2\|\mathbf{X} - \mathbf{X}^*\|_F} + \|\mathbf{X} - \mathbf{X}^*\|_F \right).
$$

Note that in order for the above bound to improve over that in Eq. (19), it indeed must in particular hold that $\|\mathbf{X} - \mathbf{X}^*\|_F < \frac{1}{2\text{rank}(\mathbf{X}^*)}$. In that case it follows that

$$
\text{RHS of (21)} \le (\mathbf{X}^* - \mathbf{X}) \bullet \nabla f(\mathbf{X}) + 5\eta\beta \sqrt{\text{rank}(\mathbf{X}^*)\|\mathbf{X} - \mathbf{X}^*\|_F}.
\tag{23}
$$

Another option, is to choose

$$
\tau_2 = \lambda_{\min}(\mathbf{X}^*), \qquad \gamma_2 = \frac{\|\mathbf{X} - \mathbf{X}^*\|_F}{\lambda_{\min}(\mathbf{X}^*)},
$$

as long as $\|\mathbf{X} - \mathbf{X}^*\|_F < \lambda_{\min}(\mathbf{X}^*)$. In this case, it holds that $\|\mathbf{X}^* \mathbf{P}_{\mathbf{X}^*, \tau}^\perp\|_F = 0$. Plugging into Eq. (22) we have that

$$
\text{RHS of (21)} \quad \le \quad (\mathbf{X}^* - \mathbf{X}) \bullet \nabla f(\mathbf{X}) + \eta\beta \|\mathbf{X} - \mathbf{X}^*\|_F \left( 2\sqrt{\text{rank}(\mathbf{X}^*)} + \frac{1}{\lambda_{\min}(\mathbf{X}^*)} \right).
$$

Note that since $\mathbf{X}^* \in \mathcal{S}_d$ it holds thst $\lambda_{\min}(\mathbf{X}^*)^{-1} \ge \text{rank}(\mathbf{X}^*)$ and thus we have that

$$
\text{RHS of (21)} \quad \le \quad (\mathbf{X}^* - \mathbf{X}) \bullet \nabla f(\mathbf{X}) + \frac{3\eta\beta \|\mathbf{X} - \mathbf{X}^*\|_F}{\lambda_{\min}(\mathbf{X}^*)}.
\tag{24}
$$

Note that here also, the above bound improves over the one in Eq. (19) only when indeed $\|\mathbf{X} - \mathbf{X}^*\|_F < \lambda_{\min}(\mathbf{X}^*)$.

Now, by using the convexity of $f$ to upper bound $(\mathbf{X}^* - \mathbf{X}) \bullet \nabla f(\mathbf{X}) \leq -(f(\mathbf{X}) - f(\mathbf{X}^*))$ and Eq. (1) to upper bound $\|\mathbf{X} - \mathbf{X}^*\|_F \leq \sqrt{\frac{2}{\alpha}(f(\mathbf{X}) - f(\mathbf{X}^*))}$ in both Eq. (23) and (24), gives the rest of the bound in the lemma.

By going through the analysis above again (basically Eq. (19) and Eq. (21)), it's clear that an $\xi$ additive error in the computation of each eigenvector $\mathbf{v}_i$ results in a single additive term $\xi$ in all of the above bounds, and hence the lemma follows. $\square$

**Lemma 7.** *[randomized update] Consider an iteration $t$ of Algorithm 2. Fix a step-size $\eta_t$ and assume that the iterate of the algorithm on this iteration is feasible and given in the following explicit form: $\mathbf{X}_t = \sum_{i=1}^{k} a_i \mathbf{x}_i \mathbf{x}_i^\top$, where each $\mathbf{x}_i$ is a unit vector, and $(a_1, ..., a_k)$ is a distribution over $[k]$. Further, suppose that each $a_i$ satisfies that $a_i \geq \eta_t/2$. Then,*

$$\mathbb{E}[h_{t+1}] \leq \left(1 - \frac{\eta_t}{2}\right) \mathbb{E}[h_t] + \frac{\eta_t^2 \beta}{2} \min\{1, \frac{5\sqrt{\sqrt{2}\mathrm{rank}(\mathbf{X}^*)}}{\alpha^{1/4}} \mathbb{E}[h_t]^{1/4}, \frac{3\sqrt{2}}{\sqrt{\alpha}\lambda_{\min}(\mathbf{X}^*)} \mathbb{E}[h_t]^{1/2}\} + \eta_t \xi_t,$$

*where $\forall t \geq 1 \; h_t := f(\mathbf{X}_t) - f(\mathbf{X}^*)$.*

*Proof.* Using the update step of Algorithm 2 we have that

$$
\begin{aligned}
h_{t+1} &= f(\mathbf{X}_{t+1}) - f(\mathbf{X}^*) = f(\mathbf{X}_t + \tilde{\eta}_t(\mathbf{v}_t \mathbf{v}_t^\top - \mathbf{x}_{i_t} \mathbf{x}_{i_t}^\top)) - f(\mathbf{X}^*) \\
&\leq f(\mathbf{X}_t) - f(\mathbf{X}^*) + \tilde{\eta}_t(\mathbf{v}_t \mathbf{v}_t^\top - \mathbf{x}_{i_t} \mathbf{x}_{i_t}^\top) \bullet \nabla f(\mathbf{X}_t) + \frac{\tilde{\eta}_t^2 \beta}{2} \|\mathbf{v}_t \mathbf{v}_t^\top - \mathbf{x}_{i_t} \mathbf{x}_{i_t}^\top\|_F^2 \\
&\leq h_t + \tilde{\eta}_t \left[ (\mathbf{v}_t \mathbf{v}_t^\top - \mathbf{x}_{i_t} \mathbf{x}_{i_t}^\top) \bullet \nabla f(\mathbf{X}_t) + \frac{\eta_t \beta}{2} \|\mathbf{v}_t \mathbf{v}_t^\top - \mathbf{x}_{i_t} \mathbf{x}_{i_t}^\top\|_F^2 \right],
\end{aligned}
$$

where the first inequality follows from the smoothness of $f$ and the second one follows since by definition, $\eta_t \geq \tilde{\eta}_t$.

By the choice of $\mathbf{v}_t$ we have that

$$(\mathbf{v}_t \mathbf{v}_t^\top - \mathbf{x}_{i_t} \mathbf{x}_{i_t}^\top) \bullet \nabla f(\mathbf{X}_t) + \frac{\eta_t \beta}{2} \|\mathbf{v}_t \mathbf{v}_t^\top - \mathbf{x}_{i_t} \mathbf{x}_{i_t}^\top\|_F^2 \leq \xi_t. \tag{25}$$

To see why this is true, observe that since $\|\mathbf{v}_t\| = \|\mathbf{x}_{i_t}\| = 1$, we have that

$$
\begin{aligned}
&(\mathbf{v}_t \mathbf{v}_t^\top - \mathbf{x}_{i_t} \mathbf{x}_{i_t}^\top) \bullet \nabla f(\mathbf{X}_t) + \frac{\eta_t \beta}{2} \|\mathbf{v}_t \mathbf{v}_t^\top - \mathbf{x}_{i_t} \mathbf{x}_{i_t}^\top\|_F^2 = \\
&(\mathbf{v}_t \mathbf{v}_t^\top - \mathbf{x}_{i_t} \mathbf{x}_{i_t}^\top) \bullet \nabla f(\mathbf{X}_t) - \eta_t \beta \mathbf{v}_t \mathbf{v}_t^\top \bullet \mathbf{x}_{i_t} \mathbf{x}_{i_t}^\top + \eta_t \beta = \\
&\mathbf{v}_t \mathbf{v}_t^\top \bullet \left( \nabla f(\mathbf{X}_t) - \eta_t \beta \mathbf{x}_{i_t} \mathbf{x}_{i_t}^\top \right) - \mathbf{x}_{i_t} \mathbf{x}_{i_t}^\top \bullet \nabla f(\mathbf{X}_t) + \eta_t \beta \leq \\
&\left( \mathbf{x}_{i_t} \mathbf{x}_{i_t}^\top \bullet \left( \nabla f(\mathbf{X}_t) - \eta_t \beta \mathbf{x}_{i_t} \mathbf{x}_{i_t}^\top \right) + \xi_t \right) - \mathbf{x}_{i_t} \mathbf{x}_{i_t}^\top \bullet \nabla f(\mathbf{X}_t) + \eta_t \beta = \xi_t,
\end{aligned}
$$

where the inequality follows from the approximated optimality of $\mathbf{v}_t$.

Thus, since by our assumption on $\{a_i\}_{i \in [k]}$ it also holds that $\tilde{\eta}_t \geq \eta_t/2$, we have using Eq. (25) that

$$
\begin{aligned}
h_{t+1} &\leq h_t + \frac{\eta_t}{2} \left[ (\mathbf{v}_t \mathbf{v}_t^\top - \mathbf{x}_{i_t} \mathbf{x}_{i_t}^\top) \bullet \nabla f(\mathbf{X}_t) + \frac{\eta_t \beta}{2} \|\mathbf{v}_t \mathbf{v}_t^\top - \mathbf{x}_{i_t} \mathbf{x}_{i_t}^\top\|_F^2 \right] + \left( \tilde{\eta}_t - \frac{\eta_t}{2} \right) \xi_t \\
&\leq h_t + \frac{\eta_t}{2} \left[ (\mathbf{v}_t \mathbf{v}_t^\top - \mathbf{x}_{i_t} \mathbf{x}_{i_t}^\top) \bullet \nabla f(\mathbf{X}_t) + \frac{\eta_t \beta}{2} \|\mathbf{v}_t \mathbf{v}_t^\top - \mathbf{x}_{i_t} \mathbf{x}_{i_t}^\top\|_F^2 \right] + \frac{\eta_t}{2} \xi_t, \tag{26}
\end{aligned}
$$

where the last inequality follows again by using $\eta_t \geq \tilde{\eta}_t$.

Taking expectation over the random choice of $i_t$ in Eq. (26), and plugging Lemma 6, we have that

$$\mathbb{E}_{i_t}[h_{t+1} \,|\, \mathbf{X}_t] \leq h_t - \frac{\eta_t}{2} h_t + \frac{\eta_t^2 \beta}{2} \min\{1, \frac{5\sqrt{\sqrt{2}\mathrm{rank}(\mathbf{X}^*)}}{\alpha^{1/4}} h_t^{1/4}, \frac{3\sqrt{2}}{\sqrt{\alpha}\lambda_{\min}(\mathbf{X}^*)} h_t^{1/2}\} + \frac{\eta_t}{2} \xi_t + \frac{\eta_t}{2} \xi_t.$$

Taking expectation over the randomness introduced on iterations $1, ..., t-1$ we have that

$$\mathbb{E}[h_{t+1}] \leq \left(1 - \frac{\eta_t}{2}\right)\mathbb{E}[h_t] + \frac{\eta_t^2\beta}{2}\min\{1, \frac{5\sqrt{\sqrt{2}\text{rank}(\mathbf{X}^*)}}{\alpha^{1/4}}\mathbb{E}[h_t^{1/4}], \frac{3\sqrt{2}}{\sqrt{\alpha}\lambda_{\min}(\mathbf{X}^*)}\mathbb{E}[h_t^{1/2}]\} + \eta_t\xi_t$$

$$\leq \left(1 - \frac{\eta_t}{2}\right)\mathbb{E}[h_t] + \frac{\eta_t^2\beta}{2}\min\{1, \frac{5\sqrt{\sqrt{2}\text{rank}(\mathbf{X}^*)}}{\alpha^{1/4}}\mathbb{E}[h_t]^{1/4}, \frac{3\sqrt{2}}{\sqrt{\alpha}\lambda_{\min}(\mathbf{X}^*)}\mathbb{E}[h_t]^{1/2}\} + \eta_t\xi_t,$$

where the first inequality follows since the function $f(x, y, z) = \min\{x, y, z\}$ is concave, and thus the inequality follows from applying Jensen's inequality. Similarly, the second inequality follows since both functions $g(x) = x^{1/4}$, $q(x) = x^{1/2}$ are also concave on $(0, \infty)$. □

## D Proof of Theorem 1

We can now turn to prove our main theorem, Theorem 1. The proof follows from deriving each one of the convergence rates in the theorem independently using the result of Lemma 7. This is done in the following Lemmas 8,9, 10. We then show that there exists a choice of step-size sequence and error-tolerance bounds for the eigenvector computations that satisfy all lemmas at once, and thus the theorem is obtained.

**Lemma 8.** *Let $C, t_0$ be non-negative scalars that satisfy:*

$$C \geq 18, \qquad \frac{C}{2} - 1 \geq t_0 \geq \frac{C}{6} - 1.$$

*Then if for all $t \geq 1$ we define $\eta_t = \frac{C}{3(t+t_0)}$, and we set $\xi_0 = \beta$ and $\forall t \geq 1 : \xi_t = \frac{\beta C}{6(t+t_0)}$, it follows that all iterates of Algorithm 2 are feasible, and*

$$\forall t \geq 1: \quad \mathbb{E}[h_t] \leq \frac{\beta C}{t + t_0}.$$

*Proof.* From Lemma 7 we have that for all $t \geq 1$,

$$\forall t \geq 1: \quad \mathbb{E}[h_{t+1}] \leq \left(1 - \frac{\eta_t}{2}\right)\mathbb{E}[h_t] + \frac{\eta_t^2\beta}{2} + \eta_t\xi_t.$$

We are going to assume throughout the proof that $\xi_t \leq \mathbb{E}[h_t]/6$. It thus follows that

$$\forall t \geq 1: \quad \mathbb{E}[h_{t+1}] \leq \left(1 - \frac{\eta_t}{3}\right)\mathbb{E}[h_t] + \frac{\eta_t^2\beta}{2}. \tag{27}$$

For all $t \geq 1$, define $v_t := \beta^{-1}\mathbb{E}[h_t]$. Dividing both sides of Eq. (27) by $\beta$, we have that

$$\forall t \geq 1: \quad v_{t+1} \leq \left(1 - \frac{\eta_t}{3}\right)v_t + \frac{\eta_t^2}{2}. \tag{28}$$

We are going to prove by induction on $t$ that $v_t \leq \frac{C}{t+t_0}$ for suitable valus of $C, t_0$ and a sequence of step-sizes $\{\eta_t\}_{t\geq 1}$. Obviously for the base case $t = 1$ to hold, we must restrict $\frac{C}{t_0+1} \geq v_1$.

Let us assume now that the induction hypothesis holds for some $t \geq 1$.

Setting $\eta_t = \frac{C}{3(t+t_0)}$ in Eq. (28) we have that

$$v_{t+1} \leq v_t\left(1 - \frac{C}{9(t+t_0)}\right) + \frac{C^2}{18(t+t_0)^2} \leq \frac{C}{t+t_0}\left(1 - \frac{C}{9(t+t_0)}\right) + \frac{C^2}{18(t+t_0)^2}$$

$$= \frac{C}{t+t_0}\left(1 - \frac{C}{18(t+t_0)}\right) = \frac{C}{t+t_0+1}\left(1 + \frac{1}{t+t_0}\right)\left(1 - \frac{C}{18(t+t_0)}\right).$$

Thus, choosing $C \geq 18$ gives:

$$v_{t+1} \leq \frac{C}{t+1+t_0}\left(1 + \frac{1}{t+t_0}\right)\left(1 - \frac{1}{t+t_0}\right) < \frac{C}{t+1+t_0}$$

as needed.

We can now set values for $C, t_0$ under the constraints that

$$i.\, C \geq 18, \qquad ii.\, \frac{C}{t_0+1} \geq v_1, \qquad iii.\, \forall t \geq 1 : \eta_t = \frac{C}{3(t+t_0)} \in [0, 2]. \qquad (29)$$

In order for our choice of step-sizes to satisfy the conditions of Observation 1, it must hold that $\{\eta_t\}_{t\geq 1} \subset [0, 2]$. Since by definition this sequence is monotonic decreasing it suffices to show it for $\eta_1$. Thus we must require that $\frac{C}{3(1+t_0)} \leq 2$, which gives us the constraint $t_0 \geq \frac{C}{6} - 1$.

It remains to deal with base case of the induction, i.e., we need to show that $v_1 = \beta^{-1}h_1 \leq \frac{C}{1+t_0}$ for our choice of $C, t_0$.

Recall that according to Algorithm 2 it holds that $\mathbf{X}_1 = \mathbf{x}_1\mathbf{x}_1^\top$, such that $\mathbf{x}_1 = \mathbf{EV}(\nabla f(\mathbf{x}_0\mathbf{x}_0^\top))$, where $\mathbf{x}_0$ is some unit vector. Using the smoothness of $f$ we have that

$$
\begin{aligned}
h_1 &= f(\mathbf{x}_1\mathbf{x}_1^\top) - f(\mathbf{X}^*) = f(\mathbf{x}_0\mathbf{x}_0^\top + \mathbf{x}_1\mathbf{x}_1^\top - \mathbf{x}_0\mathbf{x}_0^\top) - f(\mathbf{X}^*)\\
&\leq f(\mathbf{x}_0\mathbf{x}_0^\top) - f(\mathbf{X}^*) + (\mathbf{x}_1\mathbf{x}_1^\top - \mathbf{x}_0\mathbf{x}_0^\top) \bullet \nabla f(\mathbf{x}_0\mathbf{x}_0^\top) + \frac{\beta}{2}\|\mathbf{x}_1\mathbf{x}_1^\top - \mathbf{x}_0\mathbf{x}_0^\top\|_F^2\\
&\leq f(\mathbf{x}_0\mathbf{x}_0^\top) - f(\mathbf{X}^*) + (\mathbf{X}^* - \mathbf{x}_0\mathbf{x}_0^\top) \bullet \nabla f(\mathbf{x}_0\mathbf{x}_0^\top) + \frac{\beta}{2}\|\mathbf{x}_1\mathbf{x}_1^\top - \mathbf{x}_0\mathbf{x}_0^\top\|_F^2 + \xi_0\\
&\leq \frac{\beta}{2}\|\mathbf{x}_1\mathbf{x}_1^\top - \mathbf{x}_0\mathbf{x}_0^\top\|_F^2 + \xi_0 \leq \beta + \xi_0, \qquad (30)
\end{aligned}
$$

where the second inequality follows from the choice of $\mathbf{x}_1$, and the third inequality follows from the convexity of $f(\mathbf{X})$.

Setting $\xi_0 = \beta$, it follows that

$$v_1 \leq \beta^{-1} \cdot 2\beta = 2.$$

Thus we must require that $\frac{C}{1+t_0} \geq 2$, which gives us the constraint $t_0 \leq \frac{C}{2} - 1$.

Thus, the conditions in Eq. (29) boils down to the following constraints:

$$C \geq 18, \qquad \frac{C}{2} - 1 \geq t_0 \geq \frac{C}{6} - 1.$$

For $C, t_0$ that indeed satisfy these constraints we can thus conclude that

$$\forall t \geq 1 : \quad \mathbb{E}[h_t] \leq \beta v_t \leq \frac{\beta C}{t + t_0}.$$

$\square$

**Lemma 9.** *Let $C, t_0$ be non-negative scalars that satisfy:*

$$C \geq 30^{4/3}, \qquad C^{3/4} - 1 \geq t_0 \geq \frac{C^{3/4}}{6} - 1.$$

*Then if for all $t \geq 1$ we define $\eta_t = \frac{C^{3/4}}{3(t+t_0)}$, and set $\xi_0 = \beta$, $\forall t \geq 1 : \xi_t = \frac{1}{6}\left(\frac{5C^{3/4}\beta\sqrt{\sqrt{2}rank(\mathbf{X}^*)}}{\alpha^{1/4}(t+t_0)}\right)^{4/3}$, it follows that all iterates of Algorithm 2 are feasible, and*

$$\forall t \geq 1 : \quad \mathbb{E}[h_t] \leq \left(\frac{5C^{3/4}\beta\sqrt{\sqrt{2}rank(\mathbf{X}^*)}}{\alpha^{1/4}(t+t_0)}\right)^{4/3}.$$

*Proof.* From Lemma 7 we have that for all $t \geq 1$,

$$\forall t \geq 1: \quad \mathbb{E}[h_{t+1}] \leq \left(1 - \frac{\eta_t}{2}\right)\mathbb{E}[h_t] + \frac{5\eta_t^2 \beta\sqrt{\sqrt{2}\text{rank}(\mathbf{X}^*)}}{2\alpha^{1/4}}\mathbb{E}[h_t]^{1/4} + \eta_t \xi_t.$$

We are going to assume throughout the proof that $\xi_t \leq \mathbb{E}[h_t]/6$. It thus follows that

$$\forall t \geq 1: \quad \mathbb{E}[h_{t+1}] \leq \left(1 - \frac{\eta_t}{3}\right)\mathbb{E}[h_t] + \frac{5\eta_t^2 \beta\sqrt{\sqrt{2}\text{rank}(\mathbf{X}^*)}}{2\alpha^{1/4}}\mathbb{E}[h_t]^{1/4}. \tag{31}$$

For all $t \geq 1$, define $v_t := \left(\frac{5\sqrt{\sqrt{2}\text{rank}(\mathbf{X}^*)}\beta}{\alpha^{1/4}}\right)^{-4/3}\mathbb{E}[h_t]$. Dividing both sides of Eq. (31) by $\left(\frac{5\sqrt{\sqrt{2}\text{rank}(\mathbf{X}^*)}\beta}{\alpha^{1/4}}\right)^{4/3}$, we have that

$$\forall t \geq 1: \quad v_{t+1} \leq \left(1 - \frac{\eta_t}{3}\right)v_t + \frac{\eta_t^2}{2}v_t^{1/4}. \tag{32}$$

We are going to prove by induction on $t$ that $v_t \leq \frac{C}{(t+t_0)^{4/3}}$ for suitable valus of $C, t_0$ and a sequence of step-sizes $\{\eta_t\}_{t \geq 1}$. Obviously for the base case $t = 1$ to hold, we must restrict $\frac{C}{(t_0+1)^{4/3}} \geq v_1$.

Let us assume now that the induction hypothesis holds for some $t \geq 1$.

Setting $\eta_t = \frac{C^{3/4}}{3(t+t_0)}$ in Eq. (32) we have that

$$\begin{aligned}
v_{t+1} &\leq v_t\left(1 - \frac{C^{3/4}}{9(t+t_0)}\right) + \frac{C^{3/2}}{18(t+t_0)^2}v_t^{1/4}\\
&\leq \frac{C}{(t+t_0)^{4/3}}\left(1 - \frac{C^{3/4}}{9(t+t_0)}\right) + \frac{C^{7/4}}{18(t+t_0)^{7/3}}\\
&= \frac{C}{(t+t_0)^{4/3}}\left(1 - \frac{C^{3/4}}{18(t+t_0)}\right)\\
&= \frac{C}{(t+1+t_0)^{4/3}}\left(1 + \frac{1}{t+t_0}\right)^{4/3}\left(1 - \frac{C^{3/4}}{18(t+t_0)}\right)\\
&= \frac{C}{(t+1+t_0)^{4/3}}\left(1 + \frac{1}{t+t_0}\right)\left(1 + \frac{1}{t+t_0}\right)^{1/3}\left(1 - \frac{C^{3/4}}{18(t+t_0)}\right)
\end{aligned}$$

The single variable function $g(x) = x^{1/3}$ is concave on $(0, \infty)$, and thus, $g(1+x) \leq g(1)+g'(1)\cdot x = 1 + \frac{x}{3}$. Using this fact, we have that

$$\begin{aligned}
v_{t+1} &\leq \frac{C}{(t+1+t_0)^{4/3}}\left(1 + \frac{1}{t+t_0}\right)\left(1 + \frac{1}{3(t+t_0)}\right)\left(1 - \frac{C^{3/4}}{18(t+t_0)}\right)\\
&< \frac{C}{(t+1+t_0)^{4/3}}\left(1 + \frac{5}{3(t+t_0)}\right)\left(1 - \frac{C^{3/4}}{18(t+t_0)}\right).
\end{aligned}$$

Thus, choosing $C \geq (90/3)^{4/3}$ gives:

$$v_{t+1} \leq \frac{C}{(t+1+t_0)^{4/3}}\left(1 + \frac{5}{3(t+t_0)}\right)\left(1 - \frac{5}{3(t+t_0)}\right) < \frac{C}{(t+1+t_0)^{4/3}},$$

as needed.

We can now set values for $C, t_0$ under the constraints that

$$i.\, C \geq 30^{4/3}, \qquad ii.\, \frac{C}{(t_0+1)^{4/3}} \geq v_1, \qquad iii.\, \forall t \geq 1: \eta_t = \frac{C^{3/4}}{3(t+t_0)} \in [0, 2]. \tag{33}$$

As in the proof of Lemma 8 it follows that constraining $t_0 \geq \frac{C^{3/4}}{6} - 1$, will result in step-sizes that satisfy the conditions of Observation 1.

Moving to deal with the base case of the induction, again similarly to Lemma 8, we have that

$$
\begin{aligned}
v_1 &= \left( \frac{\alpha^{1/4}}{5\beta \sqrt{\sqrt{2}\mathrm{rank}(\mathbf{X}^*)}} \right)^{4/3} h_1 \leq \left( \frac{\alpha^{1/4}}{5\beta \sqrt{\sqrt{2}\mathrm{rank}(\mathbf{X}^*)}} \right)^{4/3} \cdot 2\beta \\
&= \left( \frac{\sqrt{2}\alpha^{1/4}}{5\beta^{1/4}\sqrt{\mathrm{rank}(\mathbf{X}^*)}} \right)^{4/3} < 1,
\end{aligned}
$$

where the inequality follows since $\alpha \leq \beta$. Thus we must require that $\frac{C}{(1+t_0)^{4/3}} \geq 1$, which gives us the constraint $t_0 \leq C^{3/4} - 1$.

Thus, the conditions in Eq. (33) boils down to the following constraints:

$$
C \geq 30^{4/3}, \qquad C^{3/4} - 1 \geq t_0 \geq \frac{C^{3/4}}{6} - 1.
$$

For $C, t_0$ that indeed satisfy these constraints we can thus conclude that

$$
\forall t \geq 1: \quad \mathbb{E}[h_t] \leq \left( \frac{5\beta\sqrt{\mathrm{rank}(\mathbf{X}^*)}}{2^{3/4}\alpha^{1/4}} \right)^{4/3} \quad v_t \leq \left( \frac{5C^{3/4}\beta\sqrt{\mathrm{rank}(\mathbf{X}^*)}}{2^{3/4}\alpha^{1/4}(t+t_0)} \right)^{4/3}.
$$

$\square$

**Lemma 10.** *Let $C, t_0$ be non-negative scalars that satisfy:*

$$
C \geq 2916, \qquad C^{1/2} - 1 \geq t_0 \geq \frac{C^{1/2}}{6} - 1.
$$

*Then if for all $t \geq 1$ we define $\eta_t = \frac{C^{1/2}}{3(t+t_0)}$ and $\xi_0 = \beta$, $\forall t \geq 1: \xi_t = \frac{1}{6}\left( \frac{3\sqrt{2C}\beta}{\sqrt{\alpha}\lambda_{\min}(\mathbf{X}^*)(t+t_0)} \right)^2$, it follows that all iterates of Algorithm 2 are feasible, and*

$$
\forall t \geq 1: \quad \mathbb{E}[h_t] \leq \left( \frac{3\sqrt{2C}\beta}{\sqrt{\alpha}\lambda_{\min}(\mathbf{X}^*)(t+t_0)} \right)^2.
$$

*Proof.* From Lemma 7 we have that for all $t \geq 1$,

$$
\forall t \geq 1: \quad \mathbb{E}[h_{t+1}] \leq \left( 1 - \frac{\eta_t}{2} \right) \mathbb{E}[h_t] + \frac{3\sqrt{2}\eta_t^2\beta}{2\sqrt{\alpha}\lambda_{\min}(\mathbf{X}^*)} \mathbb{E}[h_t]^{1/2} + \eta_t\xi_t.
$$

We are going to assume throughout the proof that $\xi_t \leq \mathbb{E}[h_t]/6$. It thus follows that

$$
\forall t \geq 1: \quad \mathbb{E}[h_{t+1}] \leq \left( 1 - \frac{\eta_t}{3} \right) \mathbb{E}[h_t] + \frac{3\sqrt{2}\eta_t^2\beta}{2\sqrt{\alpha}\lambda_{\min}(\mathbf{X}^*)} \mathbb{E}[h_t]^{1/2}. \tag{34}
$$

For all $t \geq 1$, define $v_t := \left( \frac{3\sqrt{2}\beta}{\sqrt{\alpha}\lambda_{\min}(\mathbf{X}^*)} \right)^{-2} \mathbb{E}[h_t]$. Dividing both sides of Eq. (31) by $\left( \frac{3\sqrt{2}\beta}{\sqrt{\alpha}\lambda_{\min}(\mathbf{X}^*)} \right)^2$, we have that

$$
\forall t \geq 1: \quad v_{t+1} \leq \left( 1 - \frac{\eta_t}{3} \right) v_t + \frac{\eta_t^2}{2} v_t^{1/2}. \tag{35}
$$

We are going to prove by induction on $t$ that $v_t \leq \frac{C}{(t+t_0)^2}$ for suitable valus of $C, t_0$ and a sequence of step-sizes $\{\eta_t\}_{t \geq 1}$. Obviously for the base case $t = 1$ to hold, we must restrict $\frac{C}{(t_0+1)^2} \geq v_1$.

Let us assume now that the induction hypothesis holds for some $t \geq 1$.

Setting $\eta_t = \frac{C^{1/2}}{3(t+t_0)}$ in Eq. (32) we have that

$$
\begin{aligned}
v_{t+1} &\leq v_t \left(1 - \frac{C^{1/2}}{9(t+t_0)}\right) + \frac{C}{18(t+t_0)^2} v_t^{1/2} \\
&\leq \frac{C}{(t+t_0)^2} \left(1 - \frac{C^{1/2}}{9(t+t_0)}\right) + \frac{C^{3/2}}{18(t+t_0)^3} \\
&= \frac{C}{(t+1+t_0)^2} \left(1 + \frac{1}{t+t_0}\right)^2 \left(1 - \frac{C^{1/2}}{18(t+t_0)}\right) \\
&\leq \frac{C}{(t+1+t_0)^2} \left(1 + \frac{3}{t+t_0}\right) \left(1 - \frac{C^{1/2}}{18(t+t_0)}\right)
\end{aligned}
$$

Thus, choosing $C \geq 2916$ gives:

$$
v_{t+1} \leq \frac{C}{(t+1+t_0)^2} \left(1 + \frac{3}{t+t_0}\right) \left(1 - \frac{3}{t+t_0}\right) < \frac{C}{(t+1+t_0)^2},
$$

as needed.

We can now set values for $C, t_0$ under the constraints that

$$
i.\, C \geq 2916 \qquad ii.\, \frac{C}{(t_0+1)^2} \geq v_1, \qquad iii.\, \forall t \geq 1 : \eta_t = \frac{C^{1/2}}{3(t+t_0)} \in [0,2]. \tag{36}
$$

As in Lemma 8, it follows that in order for our step-sizes satisfy the conditions of Observation 1, we need to require that $t_0 \geq \frac{C^{1/2}}{6} - 1$.

Also, for the base case of the induction, also similarly to Lemma 8, it holds that

$$
v_1 \leq \left(\frac{\sqrt{\alpha}\lambda_{\min}(\mathbf{X}^*)}{3\sqrt{2}\beta}\right)^2 \cdot 2\beta = \left(\frac{\sqrt{2\alpha}\lambda_{\min}(\mathbf{X}^*)}{3\sqrt{2}\sqrt{\beta}}\right)^2 < 1
$$

where the second inequality follows since $\alpha \leq \beta$ and $\lambda_{\min}(\mathbf{X}^*) \leq 1$. Thus in order to satisfy the constraint $v_1 \leq \frac{C}{(t_0+1)^2}$, it suffices to require $t_0 \leq \sqrt{C} - 1$.

Thus, the conditions in Eq. (33) boils down to the following constraints:

$$
C \geq 2916, \qquad C^{1/2} - 1 \geq t_0 \geq \frac{C^{1/2}}{6} - 1.
$$

For $C, t_0$ that indeed satisfy these constraints we can thus conclude that We can thus conclude that

$$
\forall t \geq 1 : \quad \mathbb{E}[h_t] \leq \left(\frac{3\sqrt{2}\beta}{\sqrt{\alpha}\lambda_{\min}(\mathbf{X}^*)}\right)^2 v_t \leq \left(\frac{3\sqrt{2C}\beta}{\sqrt{\alpha}\lambda_{\min}(\mathbf{X}^*)(t+t_0)}\right)^2.
$$

$\square$

We can now finally wrap-up the proof of Theorem 1.

*Proof.* The proof is an immediate consequence of Lemmas 8, 9, 10, and the observation that the step-size $\eta_t = \frac{54}{3(t+8)} = \frac{18}{t+8}$, which implicitly sets $t_0 = 8$ in all of above lemmas and corresponds to setting $C = 54$ for Lemma 8, $C = 54^{4/3}$ in Lemma 9, and $C = 2916$ in Lemma 10, satisfies all lemmas together. $\square$