[Reviews · NeurIPS 2016]

Reviewer 1

Summary

For the optimisation of an \alpha-strongly convex and \beta-smooth function over positive semidefinite matrices with unit trace, the authors present a conditional gradient method ("Frank-Wolfe") with the best available convergence rate as a function of \alpha and \beta.

Qualitative Assessment

The author seems to have found a way of reusing the "spectral" machinery Nesterov (http://epubs.siam.org/doi/abs/10.1137/100802001) introduced to the analysis of rates of convergence of randomised algorithms on smooth and strongly convex unconstrained problems to analyses of smooth and strongly convex problems over the spectrahedron. This is a very important contribution, although considering the complexity of the result, would be better suited to a journal.

Confidence in this Review

1-Less confident (might not have understood significant parts)


Reviewer 2

Summary

The paper studies the important question if the Frank-Wolfe algorithm can posses a faster then O(1/T) convergence rate for strongly convex objective when optimizing over PSD bounded trace matrices (the 'spectrahedron'), and answers this question affirmatively. The obtained new rates though are only slightly improved over the classical rate.

Qualitative Assessment

The paper studies the important question if the Frank-Wolfe algorithm can posses a faster then O(1/T) convergence rate for strongly convex objective when optimizing over PSD bounded trace matrices (the 'spectrahedron'), and answers this question affirmatively. The obtained new rates though are only slightly improved over the classical rate. The shown rates are not obtained for the original FW algorithm but for a hybrid version (Algorithm 2) which interpolates between FW and the projected gradient method. This presented variant uses a simple randomization over the previously used factors, and has almost the same iteration cost as FW. The effect of the randomized rank-1 modification is to result in an implicit regularized subproblem similar to projected gradient. The presented slight algorithm modification and used proof techniques are interesting and novel, and might open the road for further future improvements. Concern on the generality of the paper: A downside in my opinion is that the paper could be improved significantly by cutting out more clearly which properties of the used constraint set are essential to obtain the presented speedup, and discussing more if a similar speedup could also be obtained by a similar algorithm running on other sets (such as sparse vectors), not necessarily restricted to matrices. In this spirit, it would be interesting to see if this proximal idea is related to the Linear Optimization Oracle of [5]. Minor comments: - Abstract: No comma in 'an alternative, is' - Terminology 'the spectrahedron' might not be ideal terminology, as the name spectrahedron is typically used for all sets which are solutions to a linear-matrix-inequality (LMO) system, see also cone programming. - Line 11: inferior rate compared to what competitor methods? UPDATE AFTER AUTHOR FEEDBACK: I'm aware that [5,16] can be applied for polytopes. I more meant the question as to if your framework can be in any way seen as a generalization of their approach, to a wider class of sets, or if you can recover interesting polytope special cases from your framework as well. Nevertheless I thank the authors for their careful answers to the points raised by the other reviewers as well.

Confidence in this Review

3-Expert (read the paper in detail, know the area, quite certain of my opinion)


Reviewer 3

Summary

This paper proposes a randomized version of the conditional gradient method for optimizing convex functions over the spectrahedron, which involves only rank-1 updates. Rank-1 updates can be implemented by computing the largest eigenvector, so the algorithm is “projection-free”. The method described exhibits improved running times in the strongly convex-regime. From the technical point of view, the convergence proof requires a bit of care, since the randomized updates require relating the expected rank-1 step to the amount of progress in function value. This is done via a somewhat technical lemma relating decompositions into sum of rank-1 matrices for pairs of matrices which are close in Frobenius norm distance.

Qualitative Assessment

I think this is a very worthy paper, which deserves to be known. The algorithm is simple and intuitive, and the presentation is very clean. I’m not sure whether the improved convergence rate are the end of the story here. It would be nice to understand whether the upper bounds for (9) are generally tight. It seems that the algorithm does not depend on the choices of \tau and \gamma, so it would be interesting to see whether the convergence exhibited experimentally matches the provided analysis, or is significantly better. Unfortunately, experiments do not seem to be very relevant in this direction (however they do show a very good improvement compared to standard conditional gradient!) ### update ### I maintain my scores.

Confidence in this Review

2-Confident (read it all; understood it all reasonably well)


Reviewer 4

Summary

This paper extends the CG algorithm for faster optimization over a constraint set with spectrahedron. Particularly, the authors applies a randomized, rank-one modification to the classical CG algorithm. The proposed algorithm is shown to converge at rate O(1/t^2) for strongly convex objective functions. The proofs appear to be correct to the reviewer.

Qualitative Assessment

The reviewer thinks that the paper has presented some new theoretical results and the proposed algorithm has demonstrated certain improvement over classical CG. Here, the major contribution is a new randomized algorithm for accelerated CG on spectrahedron and its interesting linkage with the rank-constrained, l2-regularized linear optimization subproblem needed for CG. My major concerns are on the inconsistency between the theoretical results proven and the numerical experiments performed: 1. The analyzed results in Theorem 1 are based on a fixed step size rule of "18/(t+8)", however the experiments presented are using line search to determine the step size. Moreover, the empirical convergence rate seems to be linear. Is it possible to extend the analysis done in order to verify that? 2. The actual complexity of running the proposed algorithm should be discussed, e.g., by showing the run time of the algorithm against the benchmark algos. Especially, it seems that the EV procedure used in the current algorithm would require some custom-made power method to be executed efficiently (since it relies on the fact that the operand matrix has a structure of "sparse + rank 1"). The reviewer wonders if this will impact the real life performance of the proposed ROR-CG algorithm. 3. The proposed algorithm requires storing $X_t$ as its rank-1 decomposition, i.e., x_1,x_2,...,x_k, where (in the worst-case) the rank of $X_t$ may grow as \Theta(t). The reviewer wonders if it is memory efficient to do so, especially as the algorithm seems to require >=500 iterations to converge for large scale problem such as the movielens1m dataset. Minor comments: 1. In line 203, Eq. (8), the latter summation should be "a_j" (without "star"); and also the same correction for line 204. 2. In line 179, it should be "properties". 3. In line 178, it should be "smaller than \tau". %%%%%%%%%%%%%%%% Response to author's rebuttal %%%%%%%%%%%%%%%%%%%%%%%%%%%%%%%%%%%%%%%%% The reviewer is satisfied with the author's rebuttal. Now that as the experiments are done using the specialized power/Lanczos method, it would be interesting to see how the run-time complexity improves compared to standard CGs in a revised version of the paper. In addition, the typos / corrections raised by other reviewers should also be addressed.

Confidence in this Review

2-Confident (read it all; understood it all reasonably well)


Reviewer 5

Summary

This paper provides a new method for strongly convex smooth minimization over the spectrahedron under Linear Minimization Oracle with provable rate of convergence. The expected error of this method in some regimes is better than for the existing conditional-gradient-type methods. The performance gain of the proposed approach in comparison to existing methods is demonstrated on a problem of matrix completion.

Qualitative Assessment

Initial review. Technical quality:whether experimental methods are appropriate, proofs are sound, results are well analyzed. Besides the found errors in the proofs, I have the following comments which, I hope explain my evaluation. 1. The claim in the Lemma 1, that $b$'s form a distribution is not proved and doesn't hold. But I didn't find a place, where this claim is used in the sequel. 2. It is not clear why, for the numerical experiments, the authors choose a heuristic modification of their algorithm, despite that it doesn't have a proved convergence theorem. It would be interesting also to see how the provable method works. 3. Perhaps, the averaging over 5 independent experiments does not say much about the advantages of the new method. Novelty/originality: in any aspect of the work, theory, algorithm, applications, experimental. I find the proposed approach new and interesting. It seems that it works at least in practice. The problem is the theoretical justification. Potential impact or usefulness: could be societal, academic, or practical and should be lasting in time, affecting a large number of people and/or bridge the gap between multiple disciplines. Preliminary experiments show that this method could potentially solve practical problems faster. I also find the idea of rank-one regularization worth noting. The problem is the theoretical justification. Clarity and presentation: explanations, language and grammar, figures, graphs, tables, proper references. My overall impression is quite positive. But several places were unclear for me and should be improved. 1. It would be nice to explain, why the last identity in the first line of (6) holds. 2. It seems that in (6) the minimization is over the set of all unit vectors. But it is not stated there. 3. There are some misprints, e.g. on the line 179 "proprieties" should be "properties". 4. I could not understand, why in the long derivation between lines 346 and 347, the first inequality holds. 5. It is not clear form the provided explanation, why the bound for the first term in the last inequality in (17) holds. 6. It is not clear, why the inequality on the line 410 holds. 7. It is not clear, why the inequality on the line 418 holds. ================================================ Update after author feedback. Technical quality: I still have some doubts about the numerical experiments. The authors choose a heuristic modification of their algorithm, despite that it doesn't have a proved convergence theorem. Their explanations about line-search modification seem to be reasonable. Still, in the experiment part, they use greedy choice of the rank-one matrix instead of random choice for the provable algorithm. It is not clear, why one should do that. Novelty/originality: I find the proposed approach new and interesting. It seems that it works at least in practice. The problem is the theoretical justification. Potential impact or usefulness: Preliminary experiments show that this method could potentially solve practical problems faster. I also find the idea of rank-one regularization worth noting. The problem is the theoretical justification. Clarity and presentation: Unfortunately, there were too many places in the text, which were unclear for me. In their feedback, the authors tried to explain them, but at least the explanations for items 4 and 5, are still not clear. Also I add the points, which I considered as errors in the first version of my review. I think they should also be clarified in the text. 1. It would be nice to explain in the text, why the last identity in the first line of (6) holds. 2. It seems that in (6) the minimization is over the set of all unit vectors. But it is not stated there. 3. There are some misprints, e.g. on the line 179 "proprieties" should be "properties". 4. I could not understand, why in the long derivation between lines 346 and 347, the first inequality holds. 5. It is not clear form the provided explanation, why the bound for the first term in the last inequality in (17) holds. 6. It would be nice to explain in the text, why the inequality on the line 410 holds. 7. It would be nice to explain in the text, why the inequality on the line 418 holds. 8. The claim in Lemma 1, that $b$'s form a distribution should be deleted. 9. I think, it still should be explained in the text, why ||A*B||_F < = ||A||_{spectral}*||B||_F and why ||P_{Y,\tau}||_{spectral} \leq 1. 10. I think, it should be explained in the text, how the third inequality in (19) follows from the optimality of $w^*$. 11. The estimate (25) should also be explained more explicitly. Unfortunately, the explanation in the author's feedback did not help me to understand, how it is proved.

Confidence in this Review

2-Confident (read it all; understood it all reasonably well)


Reviewer 6

Summary

This authors proposed a modified version of the conditional gradient method on the spectrahedron with unit trace (X >= 0, tr(X) = 1), which achieves O(1/t^2) accelerated rate compared to the pervious method of O(1/t) rate, with the same complexity. The proposed method exploits the fact that, if setting X_0 to be rank one, all the following X_t at t-th iteration can be written as X_t = \sum_i a_i x_i x_i^T, where {a_i} is a probability distribution and |x_i|=1, for all i.

Qualitative Assessment

The paper is in general very interesting because it shows that a unit trace constrained PSD cone may be easier to solve than PSD cone. The proof is mostly rigorous and easy to validate. I would recommend acceptance for the paper. Other major comments: 1. (Appendix, line 355) In the projection, \|\tilde{x}_i\| might equal 0, so y_i could be undefined. How do you handle those cases? 2. So actually we need to have the parameters (\tau,\eta) satisfying \tau\gamma/(1-\gamma) >= |X-X_*| as prior. How could we get those parameters in practice? 3. (line 171) So to perform t iterations, we will have an additional O(t^2) complexity, while the original CG do not. This term is not discussed. Minor comments: 1. (line 139) Need |v|=1 in eq (6). 2. (line 117) V_t -> V in argmin 3. (Appendix eq (19)) In the last ineq, consider x_i^* = -w^*, then |w^*w^*^T-x_i x_i^*|^2_F<=4, and the constant should be 2\eta\beta instead of \eta\beta. Same problem for eq (22). 4. (Appendix line 410 and eq (22)) the rank bound is used before it is explained. == post-rebuttal comments == 1. Please explain why w*w*' \dot \gradient <= X* \dot \gradient, ideally adding a lemma in appendix. 2. There are a bunch of typo in Lemma 7, for example, 2.1 For (25), the constant should be \eta_t\beta/2 instead of \eta_t^2\beta/2. Also, please show more details why (25) holds. 2.2 All x_{i_t}x_{i_t} should be x_{i_t}x_{i_t}'.

Confidence in this Review

2-Confident (read it all; understood it all reasonably well)